# Co-activation of Akt, Nrf2, and NF-κB signals under UPR$_{ER}$ in torpid *Myotis ricketti* bats for survival

Wenjie Huang [1,6], Chen-Chung Liao[2,6], Yijie Han[3], Junyan Lv[4], Ming Lei[1], Yangyang Li[4], Qingyun Lv[4], Dong Dong[1], Shuyi Zhang[5], Yi-Husan Pan [4✉] & Jian Luo [1✉]

Bats hibernate to survive stressful conditions. Examination of whole cell and mitochondrial proteomes of the liver of *Myotis ricketti* revealed that torpid bats had endoplasmic reticulum unfolded protein response (UPR$_{ER}$), global reduction in glycolysis, enhancement of lipolysis, and selective amino acid metabolism. Compared to active bats, torpid bats had higher amounts of phosphorylated serine/threonine kinase (p-Akt) and UPR$_{ER}$ markers such as PKR-like endoplasmic reticulum kinase (PERK) and activating transcription factor 4 (ATF4). Torpid bats also had lower amounts of the complex of Kelch-like ECH-associated protein 1 (Keap1), nuclear factor erythroid 2-related factor 2 (Nrf2), and nuclear factor kappa-light-chain-enhancer of activated B cells (NF-κB) (p65)/I-κBα. Cellular redistribution of 78 kDa glucose-regulated protein (GRP78) and reduced binding between PERK and GRP78 were also seen in torpid bats. Evidence of such was not observed in fasted, cold-treated, or normal mice. These data indicated that bats activate Akt, Nrf2, and NF-κB via the PERK-ATF4 regulatory axis against endoplasmic reticulum stresses during hibernation.

[1] Shanghai Key Laboratory of Regulatory Biology, Institute of Biomedical Sciences and School of Life Sciences, East China Normal University, Shanghai 200241, China. [2] Proteomics Research Center, National Yang-Ming University, Taipei 11221, Taiwan. [3] University of Chinese Academy of Sciences, Beijing 100049, China. [4] Key Laboratory of Brain Functional Genomics (Ministry of Education and Shanghai), Institute of Brain Functional Genomics, School of Life Sciences and the Collaborative Innovation Center for Brain Science, East China Normal University, Shanghai 200062, China. [5] College of Animal Science and Veterinary Medicine, Shenyang Agricultural University, Shenyang 110866, China. [6] These authors contributed equally: Wenjie Huang, Chen-Chung Liao. ✉email: yihsuanp@gmail.com; jluo@bio.ecnu.edu.cn

Hibernation is a survival strategy used by some animals against cold winter. It consists of repeated cycles of torpor-arousal states. The torpor state usually persists for days to weeks, but the arousal state lasts less than a day. During torpor, some mammals reduce their body temperature, respiration rate, cardiac rhythm, and metabolic rate. The body temperature may drop from 37 °C to an ambient temperature as low as −2.9 °C[1], and oxygen consumption and metabolic rates may decrease to < 5% of the basal level[2]. However, these physiological depressions recover very quickly upon arousal[1,2] with no severe organ damage[3,4].

Many stresses, such as amino acid and glucose deprivation, low-temperature exposure, oxidative stress, and ischemia–reperfusion can trigger the accumulation of unfolded or misfolded proteins in the lumen of endoplasmic reticulum[5]. This phenomenon is termed endoplasmic reticulum stress. Under endoplasmic reticulum stress, mammalian cells activate the endoplasmic reticulum-associated protein degradation (ERAD) mechanism known as unfolded protein response (UPR$_{ER}$) to prevent excessive accumulation of abnormally folded proteins[6]. ERAD is achieved by exporting the proteins from endoplasmic reticulum back to the cytosol to be degraded by the ubiquitin–proteasome system[7]. During UPR$_{ER}$, cells also restore endoplasmic reticulum protein homeostasis by attenuating protein synthesis through controlling the expression of transcription factors related to ERAD, chaperone, and autophagy[5]. Another mechanism that is also activated during UPR$_{ER}$ is inhibition of apoptosis mediated by the X-box-binding protein 1 to activate the ERK1/2 signaling pathway[8]. In addition to proteostasis, UPR$_{ER}$ also controls some lipid metabolic pathways[7]. If activation of UPR$_{ER}$ and ERAD is not sufficient to overcome endoplasmic reticulum stresses, cell death pathways are activated[6,7].

During UPR$_{ER}$, the eukaryotic translation initiation factor 2α (EIF2α) is phosphorylated by the protein kinase RNA-like endoplasmic reticulum kinase (PERK)[9]. Ser49 and Ser52 are the two phosphorylation sites in EIF2α, and Ser49 phosphorylation occurs only after Ser52 is phosphorylated. When both sites are phosphorylated, protein synthesis in the cell is completely inhibited[10]. For stress adaptation[11,12], the phosphorylated EIF2α downregulates global protein synthesis but increases the translation of some proteins, such as the 78 kDa glucose-regulated protein (GRP78) and the activating transcription factor 4 (ATF4)[6,7]. GRP78 is a major endoplasmic reticulum chaperone and functions as a master regulator of UPR[13] to control the activation of transmembrane endoplasmic reticulum stress sensors (e.g., PERK, ATF6, and IRE1)[9,14]. ATF4 can turn on both protective and apoptotic signaling pathways by controlling the expression of genes involved in redox balance and amino acid metabolism[6]. Phosphorylated EIF2α also activates NF-κB[11,15,16] and modulates Akt activity[17]. Both NF-κB and Akt signaling pathways are critical for cell survival in response to various stresses[18–20]. Nrf2 signaling that plays a key role in the maintenance of redox homeostasis against aging[21] is also activated by PERK under endoplasmic reticulum stress[22].

A previous study has shown that UPR$_{ER}$ occurs in the heart of early aroused Syrian hamsters[23], the brain and brown adipose tissue of torpid thirteen-lined ground squirrels[24,25], and the skeletal muscles of torpid Daurian ground squirrels[26]. A nearly 20-fold increase in Akt abundance and activation of NF-κB were seen in the gut of hibernating squirrels[27,28]. These findings suggest that UPR$_{ER}$ is a common response of small mammals during hibernation and that NF-κB and Akt signaling pathways are activated during hibernation.

*Myotis ricketti* bats (Rickett's big-footed bat, family Vespertilionidae) torpor deeply during winter and exhibit no significant organ injury after arousal[29]. It is not clear whether torpid *Myotis*

*ricketti* bats activate UPR$_{ER}$ and survival-signaling pathways involving Akt, Nrf2, and NF-κB to protect their liver against stresses during hibernation. To investigate this possibility, we analyzed whole-cell and mitochondrial proteomes of the liver of *M. ricketti* to determine protein expression profiles in the bats at torpid, 2-h and 24-h arousal, and active states. We also used Western blotting, co-immunoprecipitation, and immunofluorescence microscopy to quantify the expression levels of several UPR$_{ER}$ indicators (i.e., GRP78, PERK, and ATF4) in torpid and active bats. We found evidence of the occurrence of UPR$_{ER}$ in torpid bats and revealed that several critical signaling pathways are coordinately activated in bats to overcome endoplasmic reticulum stress during hibernation.

## Results

**Analyses of whole-cell and mitochondrial proteomes of the liver of *M. rickettti*.** To evaluate survival adaptation involving metabolism and anti-apoptosis in torpid bats, the proteomes of liver and liver-derived mitochondria of torpid, 2-h and 24-h aroused and active bats were analyzed. As shown in Fig. 1a, 1055 and 887 proteins were identified from liver tissue and liver mitochondria, respectively, in which 213 (20%) liver proteins and 208 (23%) mitochondrial proteins were found to be differentially expressed in these bats.

With Gene ontology (GO) slim analysis, 1055 liver, 887 mitochondrial, and 628 overlapped proteins were characterized (Fig. 1b). There were 1314 non-overlapped proteins. A number of proteins identified from both samples were found to have the ability to bind RNA, metal ion, or unfolded protein (Fig. 1b and Supplementary Fig. 1b), and more mitochondrial proteins had signal transduction activity and structural role in ribosome. In the category of biological process, more liver proteins were found to be involved in small molecule metabolism, glycolytic function, negative regulation of apoptosis, and oxidative stress, while more mitochondrial proteins were involved in translation, protein folding, and TCA cycle (Fig. 1b and Supplementary Fig. 1c).

To understand protein expression profiles in various bat groups, the identified proteins of each group were examined by Principal Component Analysis (PCA) (Supplementary Fig. 2) and volcano plot analysis (Fig. 1c and Supplementary Data 1). Results of PCA showed that different bat groups were well separated except those of torpor (T) and 2-h aroused (2 h) groups, suggesting that different groups of bats had a very different protein expression profile, but the profiles of T and 2 h bats had no significant changes (Supplementary Fig. 2). The number of differentially expressed proteins including upregulated (red dots in Fig. 1c) and downregulated ones (blue dots in Fig. 1c) was increased in the following ascending order: Torpor vs. 2 h, Torpor vs. 24 h, and Torpor vs. Active. However, torpid bats had an overall lower level of protein expression than 2-h-, 24-h-aroused and active bats as seen by more dots in the areas of negative Log2 values in Fig. 1c.

Protein–protein interaction (PPI) network construction and ingenuity pathway analysis (IPA) were performed to determine the canonical pathways of identified proteins. Results showed that proteins in the hubs containing EEF2, GRP78, UBC, and PSMB6 located in the center of the PPI network were more abundant in torpid than in active bats (Fig. 2a). Results also showed that signal transduction pathways involved in EIF2, LPS/IL-1 mediated inhibition of RXR, sirtuin, fatty-acid β-oxidation I, PPARα/RXRα activation, eNOS, GP6, unfolded protein response, and PI3K/Akt were activated (Z score > 2), and that of those involved in oxidative phosphorylation, TCA cycle II (Eukaryotic), and fatty-acid α-oxidation were inhibited (Z score < 2). The pathways with unchanged protein levels were involved in mitochondrial

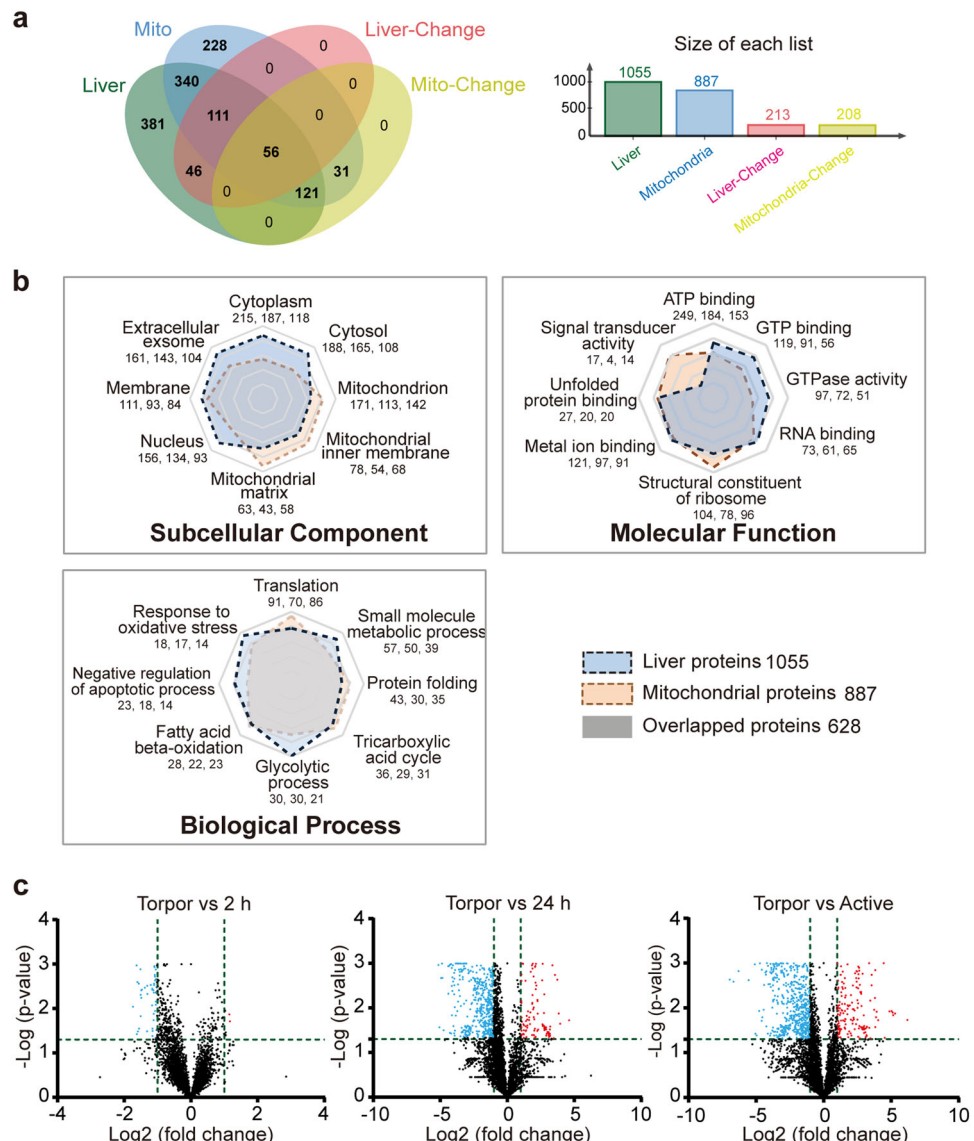

**Fig. 1 Analysis of whole-cell and mitochondrial proteomes of the liver of hibernating bats. a** Venn plots. **b** Radar plots of subcellular component, molecular function, and biological processes. The three numbers below each of the sub-category indicate the number of total, liver, and liver mitochondrial proteins identified, respectively. **c** Volcano plots: proteins that have a significantly higher level in torpid bats are dotted in red, and those that had a lower level are dotted in blue.

dysfunction and mTOR signaling (Fig. 2b and Supplementary Data 2). These results further suggest activation of UPR, EIF2, and Akt signaling but not mTOR signaling in torpid bats.

**Translational control for survival and selective expression of proteins against UPR$_{ER}$.** To investigate survival adaptation of bats during hibernation, the identified proteins involved in cell survival of each bat group were clustered by heat map (Fig. 3). Some proteins related to translation were differentially expressed among bat groups (Fig. 3a and Supplementary Data 3), including nine proteins of the 60S ribosome subunit. Among them, the levels of RPL8[a53], RPL9[a68], RPL10A[a62], RPL14[a56], RPL18[a51], and RPL22[a57] were higher in torpid than in active bats, while those of RPL4[a18], RPL19[a11], and RPLP2[a21] were lower in torpid bats. Two eukaryotic elongation factors EEF1[a40] and EEF2[a55] had a higher abundance in torpid bats. Moreover, the ribosome-binding protein 1 (RRBP1[a72]) that transports mRNAs to various subdomains

of the endoplasmic reticulum[30] and the largest RNA-binding protein vigilin (VIGLN[a70]) that is important for cell survival[31] and lipid transport had an increased abundance during torpor. Staphylococcal nuclease domain-containing protein 1 (SND1[a76]), a main component of RNA-induced silencing complex involved in miRNA processing[32,33], and the proteasome subunit beta type 4 and type 6 (PSMB4[a61] and PSMB6[a65]), normal polyubiquitin (UB[a66]), and stress-induced polyubiquitin (UBB[a67] and UBC[a69]) involved in the degradation of misfolded or damaged proteins also had a higher abundance in torpid than in active bats.

The expression levels of some chaperones, such as 78 kDa glucose-regulated protein (GRP78[a01]), heat shock protein 70 related protein 1 (HSP70.1[a35]), and protein disulfide-isomerase (PDI[a23|a37|a41]), were significantly higher in bats during hibernation. The expression levels of peptidyl-prolyl *cis–trans*-isomerase A and B (PPIA[a50] and PPIB[a36]) were also higher in bats during hibernation. These two proteins catalyze the *cis–trans*-isomerization of proline imidic peptide bonds in oligopeptides[34]. Both

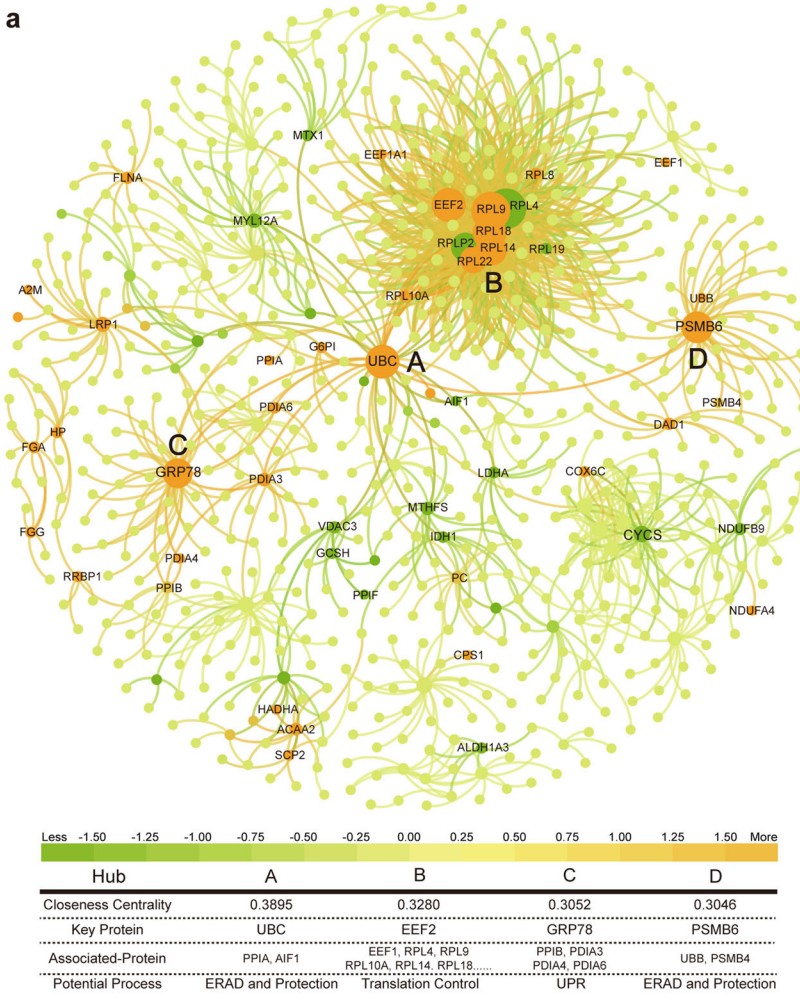

| Hub | A | B | C | D |
|---|---|---|---|---|
| Closeness Centrality | 0.3895 | 0.3280 | 0.3052 | 0.3046 |
| Key Protein | UBC | EEF2 | GRP78 | PSMB6 |
| Associated-Protein | PPIA, AIF1 | EEF1, RPL4, RPL9 RPL10A, RPL14, RPL18...... | PPIB, PDIA3 PDIA4, PDIA6 | UBB, PSMB4 |
| Potential Process | ERAD and Protection | Translation Control | UPR | ERAD and Protection |

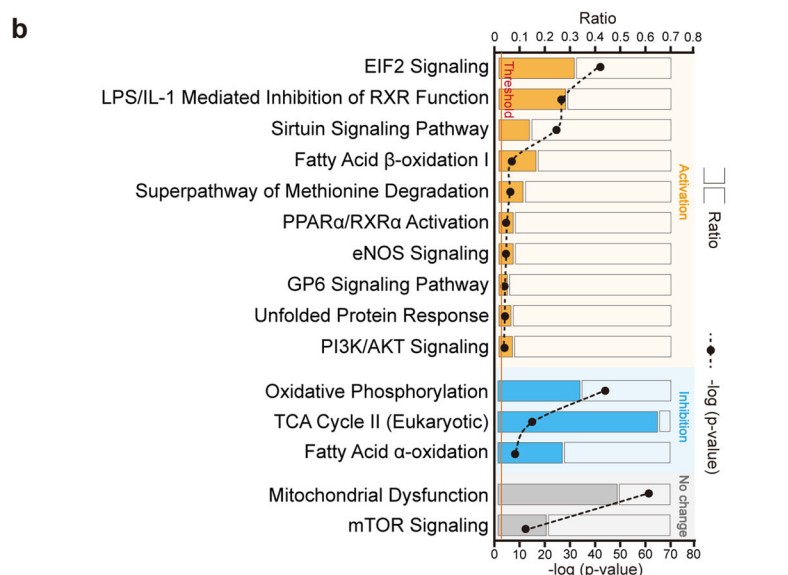

**Fig. 2 Analyses of combined whole-cell and mitochondrial proteomes of the liver of torpid and active bats. a** PPI network. Green dots or circles represent proteins with decreased levels, and orange dots or circles represent proteins with increased levels during torpor. **b** Top 15 clinical pathways predicated by IPA analysis.

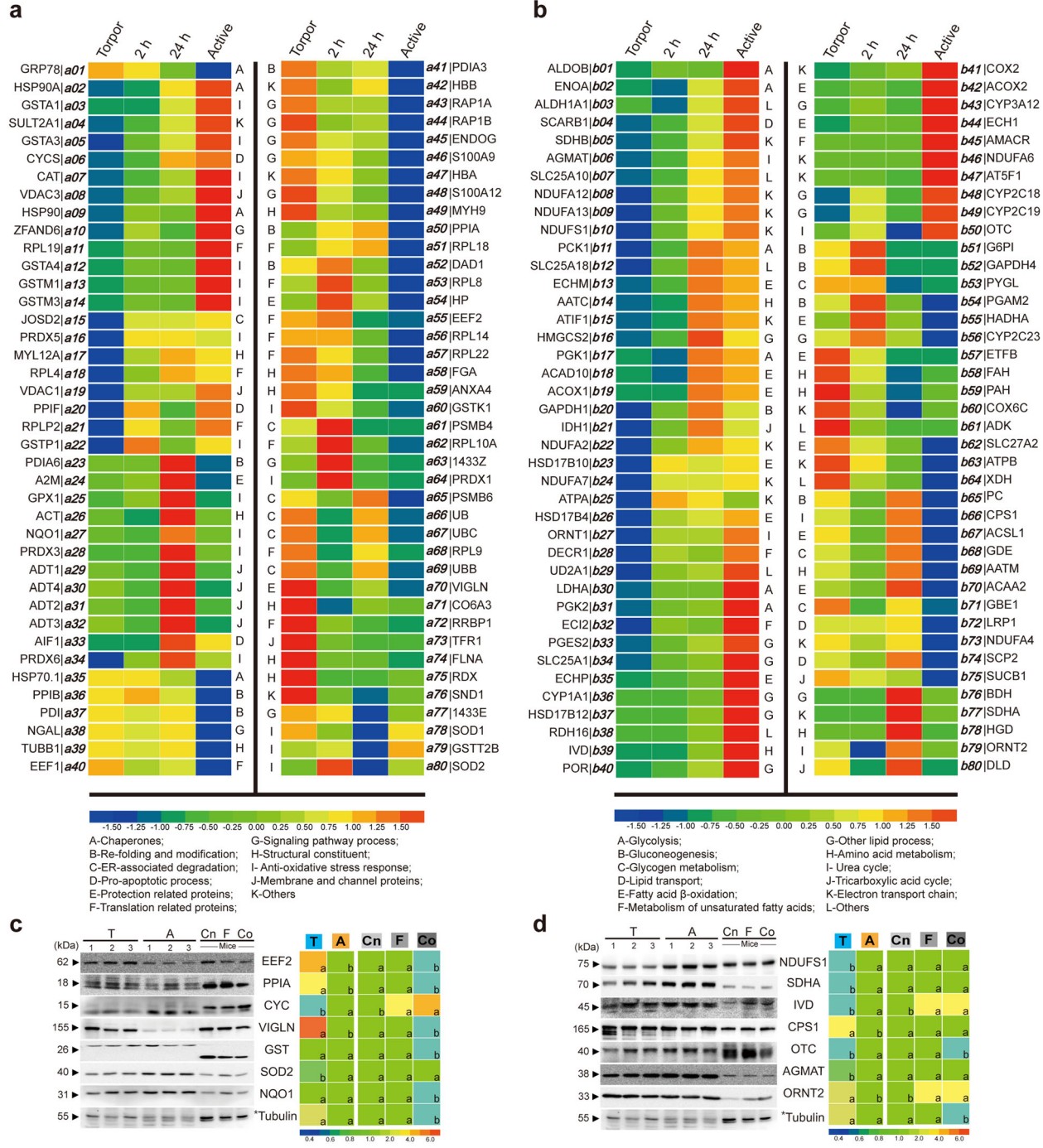

**Fig. 3 Heatmaps of identified proteins. a, b** Heatmaps of identified proteins related to cell survival (**a**) and metabolic changes (**b**). Relative expression levels are between < −1.5 and >1.5. **c, d** Results of Western blotting of proteins involved in cell survival (**a**) and metabolic changes (**b**) are also shown. Relative expression levels in bats and mice (0.4 to >6) and statistical significances (letters) were calculated. Protein levels (represented by band intensities) of torpid bats (T) were normalized to those of active bats (A), and protein levels of fasted (F) or cold-treated (Co) mice were normalized to those of control (Cn) mice. Three individuals (n = 3) from each bat group and four individuals (n = 4) from each mouse group were examined. The difference in value is significant (P < 0.05) between different letters. Asterisks indicate the same referent membrane used in this experiment. Uncropped images are shown in Supplementary Fig. 7.

PPIA and PPIB are also involved in anti-apoptosis processes[35,36]. Moreover, the defender against cell death 1 (DAD1[a52]) protein that negatively regulates programmed cell death[37] and the glutathione S-transferase kappa 1 (GSTK1[a60]) protein that downregulates adiponectin to alleviate endoplasmic reticulum stress[38,39] had a higher abundance in bats during hibernation.

**Activation of NF-kB and Akt signaling for cell survival.** Several proteins involved in NF-κB activation had a high abundance in bats during hibernation. These proteins include SND1[a76][40], S100A12 (S100A12[a48]), and the neutrophil gelatinase-associated lipocalin (NGAL[a38]). The levels of S100A9 (S100A9[a46]) and peptidyl-prolyl *cis*–*trans*-isomerase A (PPIA[a50]), which activate

both NF-κB and PI3K/Akt signaling pathways[36], were increased during torpor (Fig. 3a and Supplementary Data 3).

**Anti-apoptosis response during hibernation**. The voltage-dependent anion-selective channel (VDAC[a08|a19]) protein, which is a core component of mitochondrial permeability transition pores (MPTP), and peptidyl-prolyl *cis–trans*-isomerase F (PPIF[a20]), which is a prominent mediator of MPTP, were found to have a lower abundance in torpid than in active bats (Fig. 3a and Supplementary Data 3). The levels of both cytochrome c (CYCS[a06]) and the apoptosis-inducing factor 1 (AIF1[a33]) were also lower in torpid and 2-h arousal bats than in 24-h arousal and active bats. These results suggest that bats activate anti-apoptosis defense during hibernation.

**Utilization of lipids and ketones as energy supply during torpor**. The expression of tricarboxylate transport protein (SLC25A1[b34]) and estradiol 17-beta-dehydrogenase 12 (HSD17B12[b37]) involved in lipid production was found to be downregulated (Fig. 3b). The abundance of the nonspecific lipid-transfer protein (SCP2[b74]) that mediates the transfer of phospholipids, cholesterol, and gangliosides between membranes had a high abundance in hibernating bats. The precursor of low-density lipoprotein receptor-related protein 1 (LRP1[b72]) involved in the recycling VLDL also had a high abundance in hibernating bats, but the scavenger receptor class B member 1 (SCARB1[b04]) involved in HDL recycling had a low abundance in hibernating bats (Fig. 3b). These data suggest that bats use lipids during hibernation.

The levels of the following enzymes involved in lipid oxidation were found to be increased in torpid bats, including long-chain-fatty-acid-CoA ligase 1 (ACSL1[b67]), very long-chain acyl-CoA synthetase (SLC27A2[b62]), 3-ketoacyl-CoA thiolase (ACAA2[b70]), trifunctional enzyme subunit alpha (HADHA[b55]), and electron transfer flavoprotein subunit beta (ETFB[b57]). The levels of cytochrome P450 family proteins (CYP1A1[b36], CYP2C18[b48], 2C19[b49], and 3A12[b43]) and NADPH-cytochrome P450 reductase (POR[b40]) involved in the synthesis of lipid derivatives were decreased during torpor. These observations suggest active utilization of lipids for energy supply in bats during hibernation. Enzymes involved in ketone body generation including hydroxymethylglutaryl-CoA synthase (HMGCS2[b16]) and D-beta-hydroxybutyrate dehydrogenase (BDH[b76]) had a higher abundance in bats during hibernation, suggesting that ketone bodies are used for energy supply in torpid bats (Fig. 3b and Supplementary Data 4).

**Selective production of enzymes for amino acid metabolism**. Most enzymes involved in amino acid metabolism were maintained at a low level in bats during hibernation. However, phenylalanine-4-hydroxylase (PAH[b59]) and fumarylacetoacetase (FAH[b58]) involved in phenylalanine and tyrosine catabolism had a significantly higher level in torpid and 2-h-aroused bats than in 24-h-aroused and active bats[41]. Moreover, aspartate aminotransferase (AATM[b69]), which catalyzes the interconversion of aspartate and α-ketoglutarate to oxaloacetate and glutamate, was overexpressed during hibernation. Carbamoyl-phosphate synthase (CPS1[b66]) that produces carboxyl phosphate and ADP was also overexpressed during hibernation (Fig. 3b and Supplementary Data 4).

The levels of the following proteins identified in the proteomes were confirmed by Western blotting: EEF2[a55], PPIA[a50], NQO1[a27], SOD2[a80], GST[a13|a14|a60], VIGLN[a70], CYCS[a06] (Fig. 3c), ORNT2[b79], AGMAT[b06], OTC[b50], CPS1[b66], IVD[b39], NDUFS1[b10], and SDHA[b77] (Fig. 3d).

**Activation of UPR$_{ER}$, anti-apoptosis, and survival signals in torpid bats but not in mice**. Since GRP78 and PERK bind to each other on endoplasmic reticulum membrane and dissociate under endoplasmic reticulum stress, their expression levels in *M. ricketti* bats at torpid (T) and active (A) states and in mice of control (Cn), fasted (F), and cold-stimulated (Co) groups were determined. Results showed that the amount of PERK was significantly higher (4.4-fold) in torpid than in active bats ($P < 0.01$) but was about the same in various mouse groups (Fig. 4a). GRP78 was slightly higher in abundance in torpid than in active bats (Fig. 4a). Immunofluorescence examination revealed even distribution of GRP78 in the liver cells of active bats. In contrast, GRP78 appeared in punctate forms in the liver cells of torpid bats and in clumps in mouse liver cells (Fig. 4d). Since activation of PERK is due to the release of PERK from GRP78[14], the interaction between GRP78 and PERK was examined by co-immunoprecipitation (Co-IP). The output ratio of PERK/GRP78 (the amount of PERK pulled down by GRP78 divided by the total amount of GRP78 used in the assay) was significantly lower in torpid bats than in active bats (Fig. 4b), indicating that a lower amount of the PERK/GRP78 complex was present in torpid than in active bats. These data suggest PERK activation during torpor.

As initiation of protein translation depends on EIF2α dephosphorylation, total amounts of EIF2α and phosphorylated EIF2α (p-EIF2α on Ser49 and Ser52) were determined (Fig. 4e). Results showed that the amounts of EIF2α and Ser52 p-EIF2α had no changes between the two states of bats. However, the amount of Ser49 p-EIF2α was significantly decreased ($P < 0.05$) in torpid bats (Fig. 4e), suggesting that bats translate only certain proteins such as ATF4 during torpor as seen in Daurian ground squirrels[26]. In mice, cold-treated ones had the lowest amount of EIF2α, and the fasted ones had the highest amount of Ser52 p-EIF2α. No changes were found in the amount of Ser49 p-EIF2α in these mice (Fig. 4e). As ATF4 is a downstream effector of EIF2α signaling when the amount of active EIF2α is limited and is responsible for the expression of many stress-adaptive genes[42], its expression level was determined. Results showed that ATF4 levels were higher (4.6-fold) in torpid than in active bats (Fig. 4e) but had no change among various mouse groups, indicating activation of EIF2α signaling in torpid bats. Taken together, these data suggest that UPR$_{ER}$ occurs in bats during torpor, but not in normal, fasted, and cold-treated mice.

UPR triggers both survival and death signals under stress conditions. To exam whether UPR$_{ER}$ induces apoptosis in torpid bats, the expression levels of Bcl-2 (an anti-apoptotic protein), Bax (a pro-apoptotic protein), and the Bcl-2/Bax ratio were determined[43]. As shown in Fig. 4f, no significant difference in the levels of these proteins was found between torpid and active bats. However, cold-treated and fasted mice had a lower amount of Bcl-2 and a lower Bcl-2/Bax ratio than mice in the control group, suggesting that apoptotic response occurred in cold-treated and fasted mice.

To investigate whether anti-apoptotic response in torpid bats was correlated with ERK1/2 activation[8,43], the total amounts of ERK1/2 and phosphorylated ERK1/2 were determined. A significant higher p-ERK1/2 to ERK1/2 ratio was observed in torpid than in active bats (Fig. 4g), suggesting that ERK1/2 was activated with subsequent activation of anti-apoptosis under UPR$_{ER}$ in torpid bats. However, no sign of ERK1/2 activation was detected in cold-treated or fasted mice (Fig. 4g).

**Akt signaling pathway**. Results of Western blotting showed no changes in the levels of Akt between torpid and active bats. However, a higher amount of p-Akt (Ser473) was found in torpid

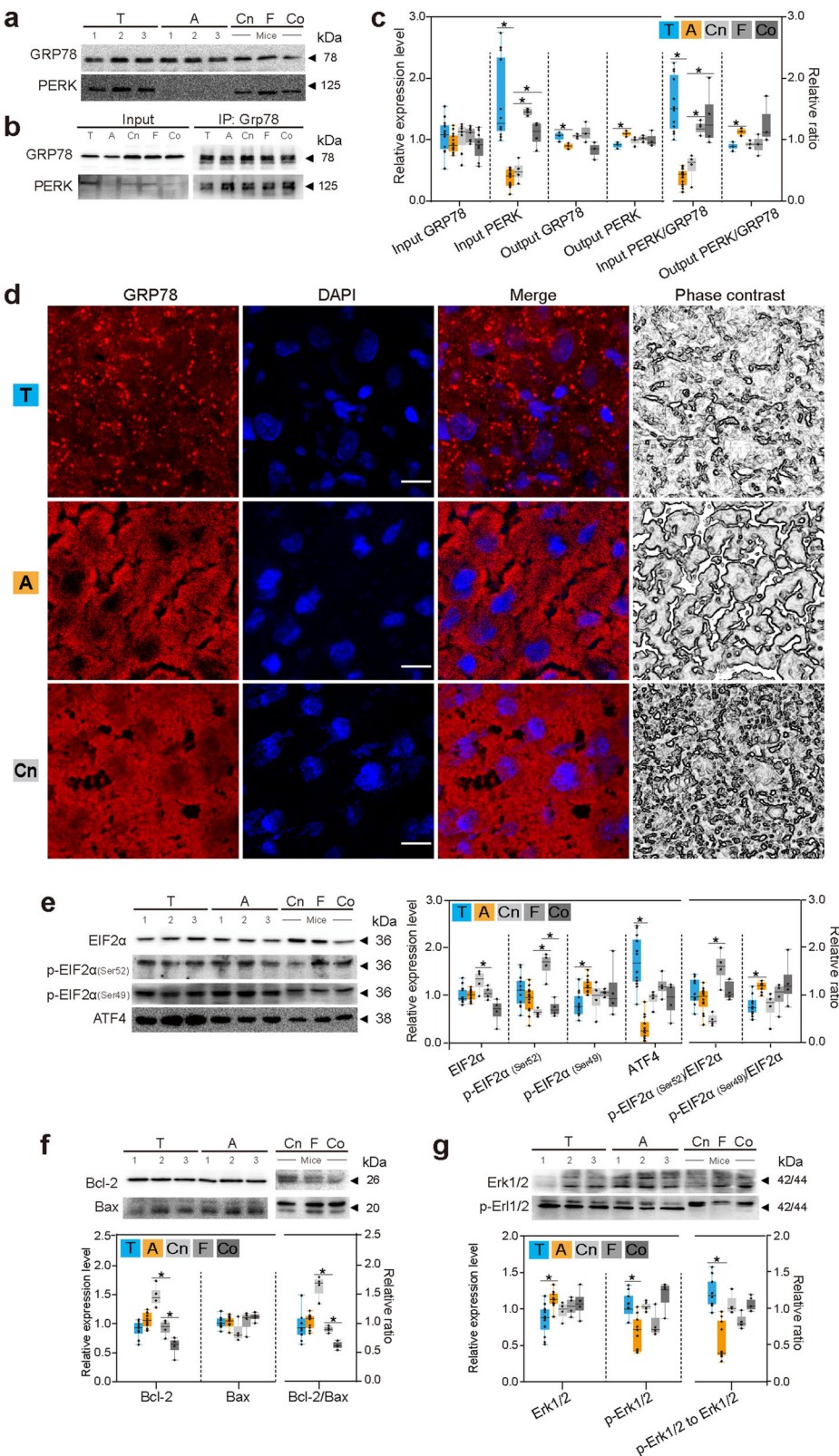

than in active bats (Fig. 5a), suggesting that Akt signaling was activated in torpid bats. This possibility was further supported by a high ratio of p-Akt (Ser473)/Akt seen in bats during torpor (Fig. 5a). In addition, cellular redistribution of GRP78 (Fig. 4d), an indication of Akt activation[13,44], was observed as described above. Among the mouse groups, cold-stimulated mice had the lowest amount of p-Akt (Ser473) and the lowest ratio of p-Akt

(Ser473)/Akt (Fig. 5a), suggesting lack of Akt activation in mice during cold treatment.

As PI3K/Akt signaling can be activated by the mTORC1 feedback pathway through EIF2α activation[17], the levels of mTOR and Ser2481-phosphorylated mTOR (p-mTOR) were determined. Results showed that the levels of both mTOR and p-mTOR were not changed in bats during torpor (Fig. 5b), implying

**Fig. 4 Quantitation of proteins involved in UPR$_{ER}$ and apoptosis. a–c** Western blotting of GRP78 and PERK (**a**) and Co-IP of GRP78 and PERK (**b**), relative amount of proteins and relative ratio of input and output PERK/GRP78 (**c**) are shown in boxplot charts. **d** Immunofluorescence images of GRP78 (in red) and nuclei (in blue), merged images, and phase-contrast images (scale bars: 20 μm). **e** Western blotting of EIF2α, p-EIF2α (Ser52), p-EIF2α (Ser49), and ATF4. **f, g** Western blotting of Bcl-2 and Bax (**f**) and Erk1/2 and p-Erk1/2 (**g**). Relative amount of proteins and relative ratio of Bcl-2/Bax, p-EIF2α (Ser52)/EIF2α, p-EIF2α (Ser49)/EIF2α, and p-Erk1/2 to Erk1/2 are shown in boxplot charts. T, A, Cn, F, and Co represent samples of torpid bats, active bats, control mice, fasted mice, and cold-treated mice, respectively. Box plots represent minimum, 25th, median (horizontal line), 75th, and maximum percentile, *$P <$ 0.05. Three individuals ($n = 3$) from each bat group and four individuals ($n = 4$) from each mouse group were examined. Uncropped images are shown in Supplementary Fig. 8.

that mTOR signaling was not involved in Akt activation in torpid bats and in cold-treated or fasted mice.

**Nrf2 signaling pathway.** Because UPR$_{ER}$ activates Nrf2 signaling in a PERK-dependent manner to overcome oxidative stresses[22], the levels of Nrf2 and its inhibitor KEAP1, which normally binds to Nrf2, were determined. No significant changes were found in the levels of both proteins (Fig. 5c). The interaction between Nrf2 and Keap1 was then examined by co-immunoprecipitation (Co-IP). Results showed that the input ratio of Keap1/Nrf2 (the amount of Keap1 divided by that of Nrf2) was constant, but the output ratio of Keap1/Nrf2 (the amount of Keap1 pulled down by Nrf2 divided by the total amount of Nrf2 used in the assay) was significantly lower in torpid than in active bats (Fig. 5d, e), suggesting a weaker or lower degree of binding between Keap1 and Nrf2 in torpid than in active bats. These data indicated that Nrf2 signaling was activated in torpid bats. However, no sign of Nrf2 activation was detected in cold-treated or fasted mice (Fig. 5e).

**NF-κB signaling pathway.** The activity of NF-κB is inhibited by its inhibitor I-κBα. The phosphorylation of I-κBα (p-I-κBα) on Ser32 causes the dissociation of NF-κB/I-κBα complex and activation of NF-κB. Western blotting results revealed that the levels of I-κBα and p-I-κBα (Ser32) were decreased, but those of NF-κB (p65) were not changed (Fig. 5f) in torpid bats. Results also showed that the ratio of p-I-κBα to I-κBα had no significant difference among various bat groups (Fig. 5h). The interaction between p65 and I-κBα was then examined by Co-IP and IF. Results showed that the input ratio of p65/I-κBα (the amount of p65 divided by that of I-κBα) was constant (Fig. 5g), but the output ratio of p65/I-κBα (the amount of p65 pulled down by I-κBα divided by the total amount of I-κBα used in the assay) was lower in torpid bats than in active bats, indicating a weaker or lower degree of interaction between p65 and I-κBα in torpid than in active bats. Since p65 was found to be largely relocated in the nuclei of liver cells in torpid bats (PC = 0.8434), but not in that of active bats (PC = 0.2324) or control mice (PC = 0.0397) (Fig. 5i), these results indicated that NF-κB signaling was activated in bats during torpor but not in mice.

## Discussion
Examination of whole-cell and mitochondrial proteomes of the liver (Figs. 1 and 2) revealed a holistic approach for survival of bats during hibernation. More liver proteins with ATP and GTP binding and GTPase activities and more mitochondrial proteins involved in signal transduction, translation, and protein folding were found in bats during torpor (Fig. 1b). Torpid bats were also found to have an overall lower protein abundance and more proteins with a significantly higher abundance (Fig. 1c and Supplementary Data 1). These findings suggest that physiological activities are greatly reduced to conserve energy during torpor.

Selective binding of 4E-binding protein 1 (4E-BP1) to EIF4E to initiate protein translation was found in golden-mantled ground

squirrels during winter hibernation[45–47]. Cellular stresses such as cold shock and hypoxia have been shown to induce both EIF2 and EEF2 signaling leading to translational reprogramming and energy conservation to ensure survival[48,49]. Endoplasmic reticulum stress and oxidative insults also induce EIF2 signaling to activate downstream activities related to cell survival such as endoplasmic reticulum-associated protein degradation (ERAD), lipid metabolism, amino acid metabolism, autophagy, and antioxidant responses[6]. Since controlling protein turnover is the most critical event in UPR$_{ER}$, many proteins are involved in this process, including chaperone GRP78, peptidyl-prolyl *cis–trans*-isomerase (PPI), protein disulfide-isomerase (PDI), polyubiquitin-C (UBC), and proteasome subunit beta type 6 (PSMB6)[6]. Therefore, the increase in the levels of EEF2, RPL, GRP78, PPI, UBC, and PSMB6 in the hubs of PPI network (Fig. 2a) is an evidence of UPR$_{ER}$ and EIF2 signaling in torpid bats (Figs. 2b and 4, and Supplementary Data 2).

In addition to the increased abundance of EEF2[a55], GRP78[a01], PDI[a23|a37|a41], PPIA[a50], PPIB[a36], UBC[a69], and PSMB6[a65] (Fig. 3a), the levels of HSP70.1[a35] that are critical for protein folding were also increased during torpor. Six of nine 60S ribosomal proteins (i.e., RPL8[a53], RPL9[a68], RPL10A[a62], RPL14[a56], RPL18[a51], and RPL22[a57]) and EEF1[a40] involved in protein translation also had a higher abundance in torpid than in active bats. Torpid bats also had a high level of PSMB4[a61], UB[a66], and UBB[a67] that are involved in protein degradation. A high level of RRBP1[a72] which is involved in ribosome-independent localization of certain mRNAs to the endoplasmic reticulum was also observed. These results suggest that bats have evolved UPR$_{ER}$ and specific signal cross talks in order to survive during hibernation.

During hibernation, torpid animals use preserved lipids as the main energy supply. Our results revealed relocation of vigilin (VIGLN[a70]) to the cell membrane in torpid bats (Fig. 3a and Supplementary Fig. 3). VIGLN is normally present in both nucleus and cytoplasm and plays a key role in secretion of very low-density lipoprotein in hepatocytes[50]. As VIGLN is also a receptor of high-density lipoprotein, which transports cholesterol to the liver, the presence of VIGLN in the cell membrane of torpid bats may be a means by which hibernating bats make better use of stored lipids. Some metabolic processes are changed in response to torpor, including a decrease in glycolysis but an increase in gluconeogenesis (Supplementary Results), use of lipids and ketones as energy supply, and selective expression of enzymes involved in amino acid metabolism, TCA cycle, and electron transport. Differential expression of proteins related to these changes may be the outcome of cross talks among signaling events involving PERK-EIF2-ATF4, Nrf2, Akt, NF-κB, and VIGLN. This possibility is supported by our finding that Nrf2, Akt, and NF-κB were coordinately activated via the PERK-EIF2-ATF4 signaling pathway in torpid *Myotis ricketti* bats (Fig. 6).

In torpid bats, we found a higher abundance of PERK, a lower amount of the PERK/GRP78 complex (Fig. 4a–c), and cellular redistribution of GRP78 (Fig. 4d), suggesting dissociation of PERK from GRP78 under endoplasmic reticulum stress. Once PERK is released from the PERK/GRP78 complex, it oligomerizes

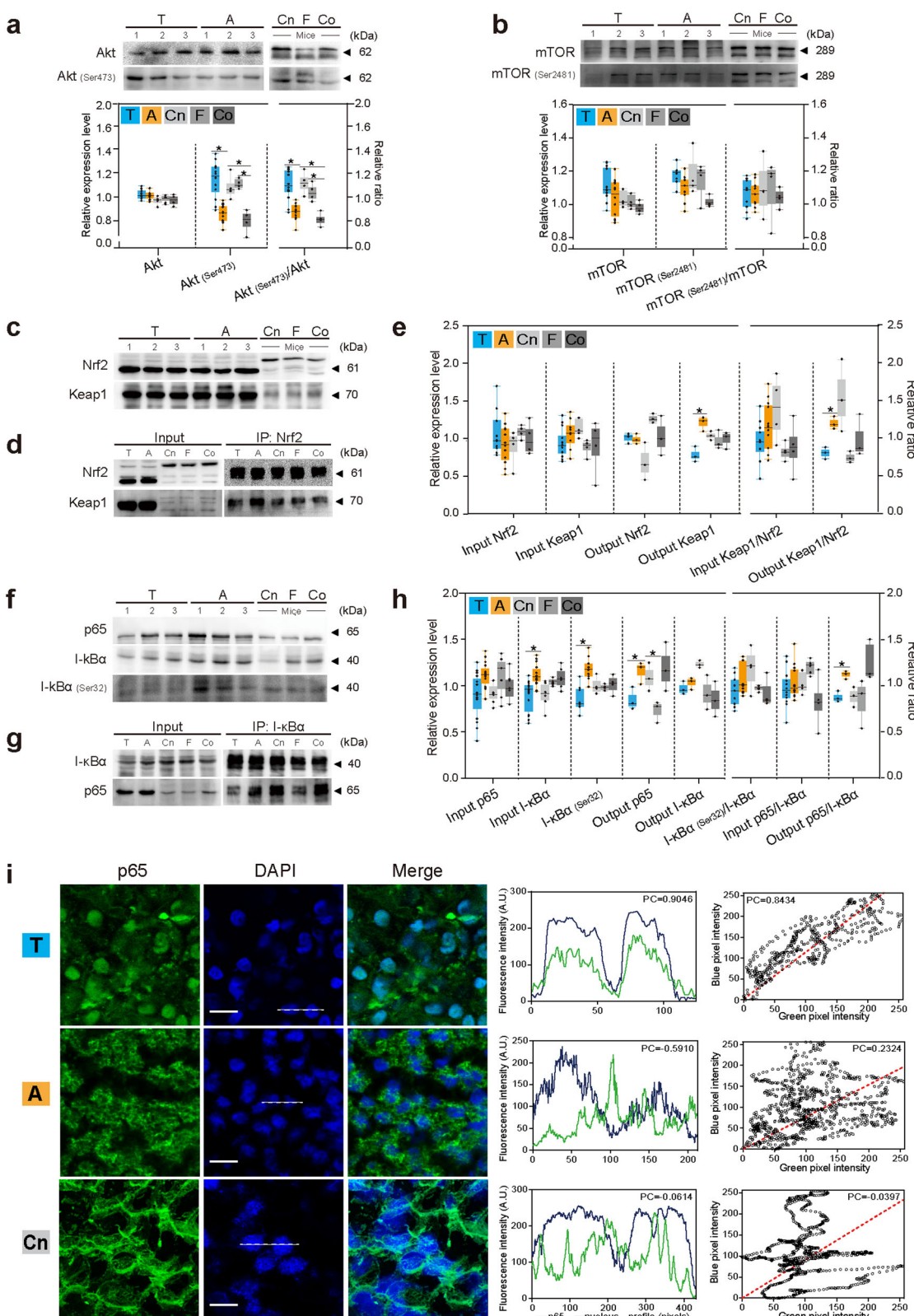

and phosphorylates EIF2α and itself, thus diminishing protein synthesis and reducing influx of proteins into the endoplasmic reticulum and alleviates endoplasmic reticulum stress[9]. Although the amounts of EIF2α and p-EIF2α (Ser52) had no significant difference in torpid and active bats, the amount of p-EIF2α (Ser49) was significantly reduced in torpid bats (Fig. 4e). Because a complete shutdown of protein synthesis occurs only when both

Ser52 and Ser49 of EIF2α are phosphorylated, Ser49 dephosphorylation, seen as reduced levels of p-EIF2α (Ser49) (Fig. 4e), would result in selective translation of some proteins[7], such as ATF4. As seen in Fig. 4e, the levels of ATF4 were found to be increased during torpor. These findings suggest that the PERK/ EIF2/ATF4 signal transduction pathway is activated in torpid bats and that these bats selectively produce some proteins to cope

**Fig. 5 Activation of Akt, Nrf2, and NF-kB. a, b** Western blotting of Akt and p-Akt (Ser473) (**a**) and mTOR and p-mTOR (Ser2481) (**b**), and relative amount of proteins and relative ratio of p-Akt (Ser473)/Akt and p-mTOR (Ser2481)/mTOR are shown in bar charts. **c–e** Western blotting of Nrf2 and Keap1 (**c**) and Co-IP of Nrf2 and Keap1 (**d**), and relative amount of proteins and relative ratio of input and output Keap1/Nrf2 (**e**). **f–h** Western blotting of p65, I-κBα, and I-κBα (Ser32) (**f**) and Co-IP of p65 and I-κBα (**g**), and relative amount of proteins and relative ratio of input and output p65/I-κBα (**h**). **i** Immunofluorescence of p65 (in green), nucleus (in blue), and overlapped areas (in purple) (scale bars: 30 μm), fluorescence intensity (A.U.) of p65, and nuclei in the areas indicated with white lines in left panels were calculated and plotted with green and blue lines, respectively. Scatterplots of blue pixel intensity versus green pixel intensity. T, A, Cn, F, and Co represent samples of torpid bats, active bats, control mice, fasted mice, and cold-treated mice, respectively. Box plots represent minimum, 25th, median (horizontal line), 75th, and maximum percentile, *$P < 0.05$. Three individuals ($n = 3$) from each bat group and four individuals ($n = 4$) from each mouse group were examined. Uncropped images are shown in Supplementary Fig. 8.

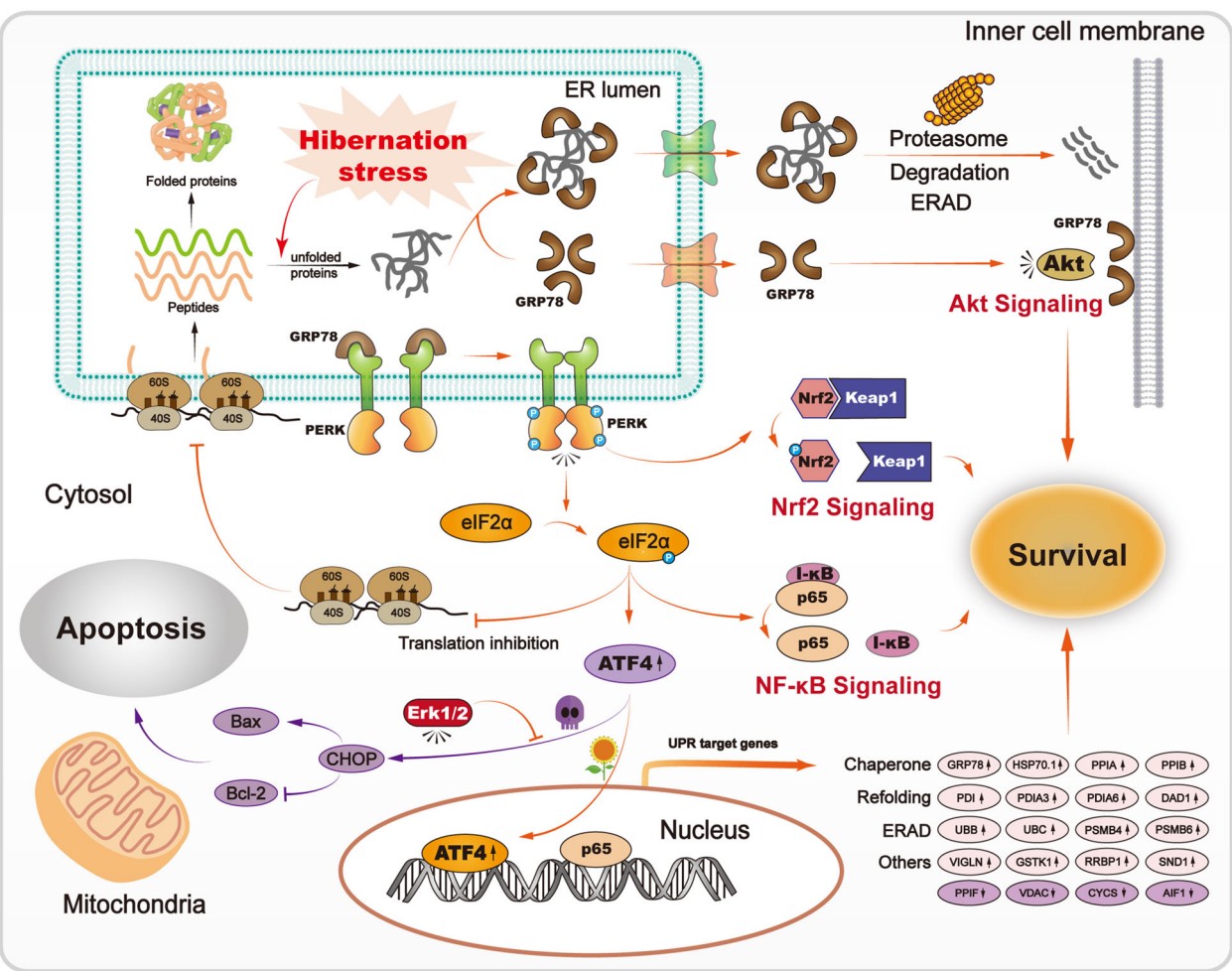

**Fig. 6 Schematic diagram of survival signals in bats under UPR$_{ER}$.** Cell death and living cells are represented by skull and sunflower, respectively. Activation (or stimulation) and inhibition are indicated by arrows and turnstile symbols, respectively.

with endoplasmic reticulum stress as seen in many hibernating mammals[51,52]. Since the levels of several UPR$_{ER}$ makers, such as ATF4, PERK, and GRP78, are increased in various tissues (e.g., brain, liver, and muscle) of different mammalian species (e.g., ground squirrels, hamsters, and bats) during hibernation[26,29,53–55], these findings also support our hypothesis that UPR$_{ER}$ is a common stress response in bats.

During endoplasmic reticulum stress, the PERK/EIF2/ATF4 signaling pathway regulates both survival and death pathways[56]. Under stress conditions, Bcl-2 promotes cell survival by inhibiting the activity of caspases, but Bax is translocated to mitochondrion membrane leading to cytochrome c (CYCS) release resulting in apoptosis. A previous study showed that apoptosis in hibernating 13-lined ground squirrels was suppressed due to a 12-fold increase in the expression of Bcl-x$_L$[46].

Since the Bcl-2/Bax ratio (Fig. 4f) was not changed in torpid bats, the activation of PERK/EIF2/ATF4 signaling during torpor is very likely intended for survival. This possibility is supported by the observation of ERK1/2 activation (Fig. 4g). As ERK1/2 plays a role in suppressing the function of pro-apoptosis proteins (e.g., DNA damage-inducible transcript 3 protein, CHOP) and enhancing the activity of anti-apoptosis proteins (e.g., Bcl-2)[8,43], it is possible that the pro-death effect of ATF4 is blocked by ERK1/2 in torpid bats. The low abundance of CYCS[a06], AIF1[a33], VDAC[a08|a19], and PPIF[a20] in torpid bats (Fig. 3a) also suggests diminished apoptosis during hibernation. The release of CYCS and AIF1 from mitochondria causes mitochondrial swelling. VDAC is the major component of the mitochondrial permeability transition pore (MPTP), and peptidyl-prolyl *cis–trans*-isomerase (PPIF[a20]) facilitates the opening of MPTP, leading to apoptosis.

The decrease in the levels of these proteins would reduce apoptosis.

Some mammals survive a cold winter by entering a hypometabolic state, such as torpor. However, long-term torpor may cause cells to die due to apoptosis or necrosis because of energy shortage. Therefore, they evolve mechanisms to preserve cells and protect organs from damage due to repeated ischemia–reperfusion in cycles of torpor-arousal. In torpid bats, both activation of PERK (Fig. 4a, b) and decreased interaction between Keap1 and Nrf2 were observed (Fig. 5d, e). Since activated PERK can phosphorylate Nrf2 under UPR$_{ER}$[57] and phosphorylated Nrf2 is released from Keap1 to trigger downstream survival signaling, these data suggest that Nrf2 is activated by PERK in bats during torpor. We also observed a higher amount of p-Akt (Ser473) and a higher ratio of p-Akt (Ser473)/Akt (Fig. 5a) as well as GRP78 redistribution in liver cells of torpid bats (Fig. 4d). As cellular redistribution of GRP78 correlates with Akt activation[58] and p-Akt (Ser473) is an active form of Akt, it is possible that Akt signaling is turned on by GRP78, which in turn activates redox mediators such as FOXOs to enhance cell survival[59].

NF-κB (p65) is activated in response to harmful stimuli (e.g., reactive oxygen species and bacterial lipopolysaccharides). The release of NF-κB (p65) from I-κBα (Fig. 5g) and the relocation of NF-κB (p65) from the cytoplasm to the nucleus (Fig. 5i) indicate NF-κB (p65) activation. As modulation of the activity of Akt[17] and NF-κB[11,15,16] is mainly mediated by phosphorylated EIF2α under stress conditions, our results suggest that Akt, Nrf2, and NF-κB work together via the PERK/EIF2/ATF4 regulatory axis (Figs. 4 and 5) to ensure cell survival under UPR$_{ER}$ (Fig. 6). The activation of NF-κB[27,60] and Nrf2[61] has been shown in hibernating ground squirrels, reflecting the importance of these proteins in small mammals during hibernation.

More evidence of coordinated activation of Akt, Nrf2, and NF-κB signal transduction pathways is that torpid bats had a higher level of S100A9[a46][62] and PPIA[a50][63]. PPIA plays a key role in anti-apoptosis response[35,36], and S100A9 regulates inflammatory and immune responses by promoting neutrophil phagocytosis via the activation of SYK, PI3K/Akt, and ERK1/2. Torpid bats also have elevated levels of S100A12[a48][62] and NGAL[a38][64] that activate NF-κB to enhance cell survival (Fig. 3a). The observation that the levels of two major anti-oxidation enzymes GST and NQO1 in torpid bats remained the same as those in active bats suggests that anti-oxidation mechanisms stay active in bats during torpor.

As mammalian hibernators undergo multiple bouts of torpor and arousal but show no significant organ damage after hibernation[1,65], they are excellent models for application of natural protective mechanisms in humans[66,67]. Fasted and cold-induced non-hibernating laboratory mice that mimic torpor-like conditions of hibernators are used to explore the molecular differences between induced and natural torpor[68–70]. To further understand the survival mechanism of mammalian torpor, we compared protein levels in mice at normal, fasted, and cold-treated states that partially mimic torpor in bats. No changes were seen in the levels of GRP78 and p-EIF2α (Ser49) (Fig. 4a, e) or in the output ratio of PERK/GRP78 in these mice (Fig. 4b). The lower level of EIF2α in cold-stimulated mice and the high level of p-EIF2α (Ser52) in fasted mice (Fig. 4e) suggest that fasting and cold stress suppress protein translation. As cold-treated and fasted mice had a significantly lower level of Bcl-2 and a lower Bcl-2/Bax ratio than control mice, liver cell apoptosis may have been triggered (Fig. 4f). Unlike torpid bats (Fig. 4g), there was no sign of ERK1/2, mTOR, Nrf2, and NF-κB activation in normal and fasted mice. These results indicate that the responses to stress of fasted and cold-treated mice were very different from that of torpid bats.

In conclusion, we have determined coordinated activation of Nrf2, Akt, and NF-κB signaling pathways via the PERK-EIF2-ATF4 regulatory axis in torpid bats (Fig. 6). We also found evidence of apoptosis inhibition by ERK1/2 and its downstream effectors (Fig. 3a). Because activation of these signal transduction pathways was not observed in fasted or cold-treated mice, it is possible that these signaling pathways are evolved to enable bats to survive a long period of hibernation. As mammalian hibernation is a complex physiological adjustment against cold and adverse surroundings, further studies using other mammals are required to learn how signal cross talks are achieved to adapt to harsh environments[71].

## Methods

**Animal ethics and tissue acquisition**. Animals used in this study are not endangered or protected species. All experiments were approved by the Animal Ethics Committee of East China Normal University (approval number AR2012/03001), *Myotis ricketti* bats were captured from Fangshan cave (39 °48′N, 115 °42′E), Beijing, China. The temperature inside the caves was ~10 °C, and the ambient temperature outside the cave was below 4 °C. Because female bats are often pregnant during hibernation season, only adult male bats were used. Twelve male torpid bats were captured using hand nets. Four of them were sacrificed immediately. Each of the remaining 8 bats was placed separately in a cloth bag and transported within 50 min to the lab, where the temperature was maintained at 28 °C. These bats were spontaneously aroused during transportation. Four bats were sacrificed 2 h after arousal, and the other 4 bats were sacrificed 24 h after arousal. Four active male bats were also captured in summer from the same cave; these bats were sacrificed immediately upon capture. The average surface and rectal temperatures were 8 and 11 °C for torpid bats, 29 and 31 °C for 2-h-aroused bats, 32 and 35 °C for 24-h-aroused bats, and 35 and 36 °C for active bats, respectively.

Twelve 11-week-old ICR male mice were obtained from Sino-British Sippr/BK Lab Animal Ltd (Shanghai, China), maintained in a 12-h dark-light cycle at 28 °C, and provided food and water ad libitum. Mice of this age were chosen as their torpor-like states had been successfully induced[68–70]. Mice were randomly divided into three groups (4 mice/group) at the seventh day after arrival: (1) ad libitum to food at 27 °C (control group), (2) ad libitum to food at 20 °C (cold-stimulated group), and (3) fasting at 27 °C (fasted group). Each condition was maintained for 2 days with water freely available[69]. Body weight, food intake, and body temperature of all mice were recorded on the first day and every 2 days (Supplementary Fig. 4 and Supplementary Data 5). All animals used in this study were sacrificed by cervical dislocation, and their livers were rapidly removed, snap-frozen in liquid nitrogen, and stored at −80 °C until used.

**Protein sample preparations**. Liver proteins were obtained from 0.1 g of liver tissue by homogenization in a tube containing ceramic beads and 1 mL lysis buffer (10% glycerol, 2% SDS, 1.25% β-mercaptoethanol, 25 mM Tris-HCl, pH 6.8, 12.5 mM EDTA, 1/50 Tablet of cOmplete™ EDTA-free protease inhibitor cocktail). Liver homogenates were boiled at 100 °C for 10 min and then centrifuged at 12,000 × *g*, 4 °C for 15 min. Liver mitochondria were extracted from 0.02 g liver tissue using the Mitochondria Isolation Kit for Tissue (89801, Thermo Scientific, USA). Mitochondrial proteins were obtained by boiling isolated mitochondria in 0.2 mL lysis buffer at 100 °C for 10 min and collecting supernatants after centrifugation at 12,000 × *g*, 4 °C for 15 min. Several mitochondrial protein markers (e.g., NADH-ubiquinone oxidoreductase 75 kDa subunit and glutamate dehydrogenase 1) were checked to ensure the purity of mitochondrial fractions (Supplementary Fig. 5). All protein supernatants were divided into small aliquots and stored at −80 °C. One aliquot was used to determine protein concentration using the Quick Start™ Bradford protein assay kit (Bio-Rad, USA).

**Protein recovery and mass spectrometric analysis**. Each protein sample was fractionated by 10% SDS-PAGE (0.5 µg/lane), and the gel was stained for 10 min with Coomassie Brilliant Blue G-250 (Bio-Rad, Hercules, CA). After washing with distilled water, each lane of the gel was sliced into ten equal pieces. Each gel piece was placed in a 2-mL tube and destained with 25 mM NH$_4$HCO$_3$ and 50% (v/v) acetonitrile (1:1). After drying in a Speed-Vac (Thermo Electron, Waltham, MA), gel pieces were incubated with 25 mM NH$_4$HCO$_3$ and 1% β-mercaptoethanol in dark for 20 min followed by an incubation in a solution containing 5% 4-vinylpyridine in 25 mM NH$_4$HCO$_3$ and 50% acetonitrile (1:1) for 20 min. Proteins in each gel piece were then digested with modified trypsin (Promega, Mannheim, Germany) in 25 mM NH$_4$HCO$_3$ (1%, w/v) at 37 °C overnight. The tryptic peptides were extracted with 25 mM NH$_4$HCO$_3$ for 10 min, dried, and stored at −20 °C until used. This gel-based protein fractionation and recovery method was used because all the techniques involved are well-established in our lab.

Tryptic peptides were then dissolved in 10 µL formic acid (0.1%, v/v) and loaded onto an online nanoAcquity ultra Performance LC system (Waters, Manchester, UK), which was coupled to a linear ion trap-orbitrap hybrid (LTQ-Orbitrap XL™) mass spectrometer (Thermo Scientific, San Jose, CA). The mobile

phases were solution A (0.1% formic acid in water) and solution B (0.1% formic acid in acetonitrile). Peptides loaded were first captured and desalted in a C18 PepMap trap column (180 μm inner diameter, 20 mm length, 5-μm beads, 100 Å pore size) and then chromatographed in a C18 tip column (13.5 cm length, 75 μm inner diameter, and 5-μm beads; YMC-Gel) with 5–35% linear solution B gradient for 90 min, followed by 35–95% solution B gradient for 10 min at a flow rate of 0.5 μL/min. Eluted peptides were ionized with a spray voltage of 2 kV and introduced into the mass spectrometer. Mass spectrometry data were recorded in a data-dependent acquisition mode (isolation width: 1.5 Da), in which one full MS survey scan ($m/z$: 200–1500) at high resolution (> 30,000 full width at half maximum) was followed by MS/MS scan of the six most intensely charged ions ($2^+$ and $3^+$). Fragmented ions of each selected precursor were generated by collision-induced dissociation with helium gas at 35% collision energy (Supplementary Fig. 6).

**Protein identification**. The Xcalibur software package (version 2.0.7 SR1, Thermo-Finnigan Inc., San Jose, CA) was used to process the RAW data and the data were analyzed by PEAKS software (v. 8.5, Bioinformatics Solutions Inc., Waterloo, Ontario, Canada) and searched for the best-matched peptides in our bat protein database (containing 539,834 entries)[29]. All proteins identified were annotated with UniPort ID[29]. The following parameters were used to analyze proteins: 20 ppm peptide mass tolerance and 0.8 Da fragment mass tolerance; precursor mass search type, monoisotopic; enzyme, trypsin; max missed cleavage, 2; nonspecific cleavage, 0; fixed modification; S-pyridylethylation; variable modification, methionine oxidation; and variable PTMs per peptide, 2. Results were adjusted to 0.5% FDR at peptide spectrum matches, $-10 \log P > 20$, unique peptides ≥ 1, and de novo ALC score ≥ 80%. For quantitation, MS spectral counts were normalized to the sum of the spectral counts of a sample[72] (Supplementary Fig. 6). Information on identified peptides and their charge states has been submitted to ProteomeXchange (https://doi.org/10.6019/PXD016109 containing 160 Raw, 160 Mgf, and 1 MZID files)[73].

**Classification and functional analysis of proteins**. Non-overlapped proteins (Supplementary Data 6) were classified by Gene Ontology (GO) according to their cellular component, molecular function, and biological process (Fig. 1b). The functional networks were created by NetworkAnalyst and processed with Gephi. The closeness centrality (CC) of highly connected proteins was calculated, and the top 4 hubs (CC > 0.3) in the network are presented in Fig. 2 and Supplementary Data 7. The clinical pathways of identified proteins were determined by INGE-NUITY® Pathway Analysis (IPA) based on significant relationships among the proteins. Statistical significance was represented by $P$ value of overlap calculated with Fisher's exact test. The maximum false discovery rate of the pathway was <5% when the $P$-value threshold was <0.05. Proteins related to survival or metabolism were selected based on GO annotation and heatmaps[74] (Fig. 3, Supplementary Data 3 and 4).

**Immunoblotting assay**. Equal amounts of total proteins from torpid and active bats ($n = 3$, three individuals from each group) and control, fasted, and cold-treated mice ($n = 4$, four individuals from each group) were separated by SDS-PAGE and transferred onto a 0.2-μm PVDF membrane (Millipore, USA) using an electro-blotting apparatus[75]. Each PVDF membrane was blocked in TBST blocking buffer and then probed with an appropriate antibody (Supplementary Data 8) that recognizes conserved epitopes of proteins across several mammalian species. After washing, the blots were reacted with appropriate secondary antibodies and visualized using a chemiluminescent HRP kit (Millipore, USA). Images were captured with ImageQuant™ LAS-4000 (Amersham Biosciences, USA), and all protein bands detected were quantified using the ImageQuant™ TL software (version 7.0, Amersham Biosciences, USA). The relative quantity of a protein was determined by dividing the density value of the protein band on western blot by the total density value of all protein bands in a lane of a Ponceau stained blot[76] (Supplementary Figs. 7 and 8). Four independent runs of western blotting of the sample from each bat were performed. Statistical significances between torpid and active states of bats were determined by student's $t$-test. Statistical significances among the three mouse groups were determined by one-way ANOVA with post hoc Holm–Sidak test (Supplementary Data 9). A $P$ value < 0.05 was considered significant.

**Protein co-immunoprecipitation**. Protein samples were prepared separately from torpid and active bats and control, fasted, and cold-treated mice ($n = 3$, three individuals from each group). Co-IP was performed using the Immunoprecipitation Kit from Thermo Fisher Scientific (Catalog no. 10006D). Briefly, Dynabeads were incubated with a specific antibody (Ab) for 30 min at 27 ℃, and the bead-Ab complexes were washed with Ab binding and washing buffer (PBS, pH 7.4, with 0.01% Tween-20) and then incubated with the protein sample (liver homogenate), referred to as input protein sample, for 30 min at 27 ℃. The bead-Ab-Ag complexes were washed three times with washing buffer, eluted with elution buffer (50 mM Tris-HCl, pH 6.8, 10% glycerol, 1.25% β-mercaptoethanol, 2% SDS, 0.1% bromophenol blue), and heated at 100 ℃ for 10 min. The eluted protein is referred to as output protein sample. Both input and output protein samples were further analyzed by western blotting. Statistical significances between torpid and active states of bats were determined by student's $t$-test, and those among the three mouse

groups were determined by one-way ANOVA with post hoc Holm–Sidak test (Supplementary Data 9). A $P$ value < 0.05 was considered significant.

**Immunofluorescence microscopy**. Liver tissues were cut into 0.5-cm cubes, embedded in O.C.T. (optimal cutting temperature) compound (Miles Scientific), and frozen in liquid nitrogen. Each 5-μm-thick cryostat section was placed on a glass slide, air-dried, and incubated with blocking solution (1% BSA and 0.3% TritonX-100) for 2 h. Sections were then incubated with a specific primary antibody overnight at 4 ℃, followed by a secondary antibody for 1 h at room temperature. Nuclear DNA was stained with 4′,6-diamidino-2-phenylindole (DAPI). Sections were then examined using a laser scanning Olympus FluoView FV10i confocal microscope (Tokyo, Japan). Double fluorescence from green and red channels was imaged using an excitation wavelength of 547 and 645 nm. For analysis of overlapping, images were analyzed using the JACoP Plugin of ImageJ to determine the intensity of each pixel in corresponding channels and their Pearson's correlation coefficient (PC). A PC > 0.7 was considered as positive, strong correlation, and high degree of image overlap.

**Statistics and reproducibility**. Four bat groups, including torpor, 2-h- and 24-h arousal, and summer activation, were assayed. Each group contained four bats. As both whole-cell and mitochondrial proteomes of the liver of each bat were analyzed, a total of 32 proteomic data sets were generated. Proteomes of liver and liver-derived mitochondria of each bat were compared (Fig. 1a, b). The proteomic data of all bat groups were also compared (Figs. 1c, 2, and 3). Unless otherwise specified, statistical significances were determined by student's $t$-test (between the two bat groups) and by one-way ANOVA (among the three mouse groups) with post hoc Holm–Sidak test. A $P$ value < 0.05 was considered significant.

**Reporting summary**. Further information on experimental design is available in the Nature Research Reporting Summary linked to this paper.

## Data availability

The mass spectrometry proteomics data of this study have been deposited to the ProteomeXchange Consortium via the PRIDE[77] partner repository with the dataset identifier PXD016109. All other data are available from the corresponding author on reasonable request.

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

## Acknowledgements

We thank Dr. Chao-Hung Lee for editing the manuscript and providing valuable advice. We also thank professor Xiao-Bing Yuan for providing experimental materials. This study was funded by the National Science Foundation of China (No. 31100273) to Yi-Hsuan Pan, the grant Aim for the Top University Plan (No. 104AC-D101) from Taiwan Ministry of Education to Chen-Chung Liao, and the National Key Research and Development Program of China (2018YFC1105100) to Jian Luo.

## Author contributions

W.J.H., Y.H.P., C.C.L., D.D., and M.L. generated and analyzed data. W.J.H. and Y.H.P. generated figures. W.J.H, Y.J.H., C.C.L., J.Y.L., Q.Y.L., and Y.Y.L. performed experiments. Y.H.P., C.C.L., S.Y.Z., and J.L. provided experimental materials. Y.H.P. and W.J.H. designed the study and wrote the manuscript.

## Competing interests

The authors declare no competing interests.
