## [Peer Review File · Communications Biology]

Reviewers' comments:

Reviewer #1 (Remarks to the Author):

This is a comprehensive assessment of several signaling nodes relevant to hibernation in bats. The authors also follow up proteomics approaches with western blots and Co-IP to add depth to the paper. Overall, the paper warrant publication and I provide detailed comments below:

- The title is very cumbersome to read, please consider revising

- Abstract

o Acronyms should be written out completely the first time they appear in the text, check the journals policy on acronyms in the abstract

o Line 52 "hash" should be harsh

o Line 53 "liver and liver" would read better if "liver and liver-derived mitochondria"

o Its not clear after reading the abstract what was done – you mention "proteomes" were measured, yet only the results of western blots are outlined?

o Related to the above comment - these two sentences need to be revised to help the reader better understand what was done: "Using PPI network and IPA analyses, we examined the proteomes of liver and liver mitochondria of hibernating *Myotis ricketti* bats and found UPRER-related survival adaptation and a global metabolic adjustment in torpid bats. We investigated whether torpid bats are under UPRER and activate Akt, Nrf2, and NF- κ B signaling pathways that are critical for cell survival." Actually, the last sentences of the introduction (line 124 to 131) do a much better job of describing to the reader what was done.

- Introduction

o Why on line 86 do you specifically state that hibernation may aid subnormothemic preservation approaches?

o Line 88 "Stresses" should not be capitalized

- Experimental procedures

o Why were only male bats used?

o Are 2h versus 24 h aroused bats fully active and awake? Some animals which "arouse" in the middle of hibernation are not actually active– while there Tb is returned to normal they are still not moving around.

o Why were 11-week old mice used? In general, would be good to outline the relevance of including mice and how when compared to hibernation experimental conditions, it offers a different perspective, even though this is fairly obvious

o What was the body temperature of the mice monitored?

o What controls were performed to ensure the purity of the mitochondrial fraction?

o Why did the authors choose SDS-PAGE to fractionate proteins prior to mass spec, instead of other methods (e.g. on column methods)? How does recovery of proteins from gels compare to other methods.

o During Co-IP how efficient is the elution process? Does any antibody remain bound to its antigen?
Results

o Could consider labeling the different parts of Figure 1B with i, ii, iii – this way you don't have to refer to it in the text by "upper right". I would also consider adding the title "subcellular, molecular function, and biological process" inside its respective grey box. Similar comments for other figures.

- Discussion

o Many of the proteins measured are followed by a superscript – e.g. see line 519 which lists a superscript number for each protein listed. What are these in reference to? These are easily confused with references.

o Since the amount of data and discussion is so dense, it would be good to introduce figure 6 earlier and use it as a guide for the reader as you describe results.

Reviewer #2 (Remarks to the Author):

General comments:

This manuscript describes a significant proteomics study examining factors involved in stress in hibernating and active bats. Although the findings are interesting, they are not especially novel since stress proteins have been found to increase in a number of hibernators. I was disappointed that the authors did not cite some of these studies (by Hannah Carey, Sandy Martin, Frank van Breukelen, and others) in the discussion. These labs have studied stress proteins, NF- κ B, apoptosis and elongation factors in hibernators and published several papers each. The lack of citations in this instance, especially given the similarities in the current authors' findings seems strange. Please do a thorough review of the hibernation literature and discuss the similarities between these studies and the current one.

Beyond this general comment, I have some suggestions for revisions that should be addressed:

- Throughout the manuscript, the bat is referred to by its scientific name only. What is the common name for this animal? Please include this information at least once.
- For the immunoblotting, how much protein was loaded for each type of lysate (in ug)? Also, I'm not sure why the developed band was normalized to the Ponceau band of the same size. This seems like an odd way to normalize the data. There could be a lot of other proteins near the one of interest that affect the Ponceau staining. This could skew the results significantly.
- The manuscript needs a stronger rationale for the mouse experiments. Why were they done? Certainly, there are enough studies out there about cold exposure in mice and everyone can agree that cold exposure in mice is not the same as torpor in a hibernator. I don't really see what these experiments add beyond saying this same thing.
- At the top of page 10 (line 448) the authors state: "To investigate whether anti-apoptosis in torpid bats was mediated by Erk1/2 signaling..." The experiments done do not show any sort of mediation, however. To do this, one would have to block Erk1/2 signaling and show that the effect goes away. This should be reworded.
- Also, on page 13 (line 513), the authors state that "...we have determined for the first time the cross talk among Nrf2, Akt and NF- κ B signaling...". Although the IP assays did show interaction between some of these proteins, I think that this is too strong a statement. The study did not really determine the cross talk but it did show the possibility.

Reviewer #1 (Remarks to the Author):

--General comments

This is a comprehensive assessment of several signaling nodes relevant to hibernation in bats. The authors also follow up proteomics approaches with western blots and Co-IP to add depth to the paper. Overall, the paper warrant publication and I provide detailed comments below:

--Advice

--Title

1. The title is very cumbersome to read, please consider revising.

Response: We have revised the title as “Co-activation of Akt, Nrf2, and NF- κ B signals under UPR_{ER} in torpid *Myotis ricketti* bats for survival”.

--Abstract

2. Acronyms should be written out completely the first time they appear in the text, check the journals policy on acronyms in the abstract.

Response: Accordingly, we have carefully rewritten the entire section of abstract to fit the reviewer’s requests and official word limit of **150** words as follows (**Lines 52-63**):

Main Text: Lines 52-63

Bats hibernate to survive stressful conditions. Examination of liver cellular and mitochondrial proteomes of *Myotis ricketti* revealed that torpid bats had endoplasmic reticulum unfolded protein response (UPR_{ER}), global reduction in glycolysis, enhancement of lipolysis, and selective amino acid metabolism. Compared to active bats, torpid bats had higher amounts of phosphorylated serine/threonine kinase (p-Akt) and UPR_{ER} markers such as PKR-like ER kinase (PERK) and activating transcription factor 4 (ATF4) and lower amounts of the complex of Kelch-like ECH-associated protein 1 (Keap1), nuclear factor erythroid 2-related factor 2 (Nrf2), and nuclear factor kappa-light-chain-enhancer of activated B cells (NF- κ B) (p65)/I- κ B α . Cellular redistribution of 78 kDa glucose-regulated protein (GRP78) and reduced binding between PERK and GRP78 were also seen in torpid bats. Evidence of such was not observed in fasted, cold-treated, or normal mice. These data indicated that bats activate Akt, Nrf2, and NF- κ B via the PERK-ATF4 regulatory axis against ER stresses during hibernation.

3. Line 52 “hash” should be harsh.

Response: This typing error has been substituted by the new abstract.

4. Line 53 “liver and liver” would read better if “liver and liver-derived mitochondria”.

Response: We seriously reconsidered the wording and revised it using “liver and liver-derived mitochondria” (Lines 214-215, 275) or “liver cellular and mitochondrial” (Lines 52, 117-118, 213, 273, 853, and 866) at where it is appropriate.

5. Its not clear after reading the abstract what was done – you mention “proteomes” were measured, yet only the results of western blots are outlined?

6. Related to the above comment - these two sentences need to be revised to help the reader better understand what was done: “Using PPI network and IPA analyses, we examined the proteomes of liver and liver mitochondria of hibernating *Myotis ricketti* bats and found UPR_{ER}-related survival adaptation and a global metabolic adjustment in torpid bats. We investigated whether torpid bats are under UPR_{ER} and activate Akt, Nrf2, and NF-κB signaling pathways that are critical for cell survival.” Actually, the last sentences of the introduction (line 124 to 131) do a much better job of describing to the reader what was done.

Response: Totally agree! We have rewritten the entire section of abstract to fit the reviewer’s requests and official word limit of 150 words (Lines 52-63).

Main Text: Lines 52-63

Bats hibernate to survive stressful conditions. Examination of liver cellular and mitochondrial proteomes of *Myotis ricketti* revealed that torpid bats had endoplasmic reticulum unfolded protein response (UPR_{ER}), global reduction in glycolysis, enhancement of lipolysis, and selective amino acid metabolism. Compared to active bats, torpid bats had higher amounts of phosphorylated serine/threonine kinase (p-Akt) and UPR_{ER} markers such as PKR-like ER kinase (PERK) and activating transcription factor 4 (ATF4) and lower amounts of the complex of Kelch-like ECH-associated protein 1 (Keap1), nuclear factor erythroid 2-related factor 2 (Nrf2), and nuclear factor kappa-light-chain-enhancer of activated B cells (NF-κB) (p65)/I-κBα. Cellular redistribution of 78 kDa glucose-regulated protein (GRP78) and reduced binding between PERK and GRP78 were also seen in torpid bats. Evidence of such was not observed in fasted, cold-treated, or normal mice. These data indicated that bats activate Akt, Nrf2, and NF-κB via the PERK-ATF4 regulatory axis against ER stresses during hibernation.

--Introduction

7. Why on line 86 do you specifically state that hibernation may aid subnormothemic preservation approaches?

Response: To avoid confusions and arguments, and also to reach the minimum word limit of 5000 (Introduction, Results, and Discussion) we have deleted the entire sentence.

♥ Explain a little bit more here:

Mammalian metabolites such as hydrogen sulfide (H₂S) and 5-adenosine monophosphate (5-AMP) is previously found to elicit physiological effects similar to hibernation in those non-hibernators. Since H₂S supplementation may be a novel method for successful organ preservation at subnormothermic temperatures, we mistakenly assumed the relationship between molecular mechanism of hibernation and organ preservation at subnormothermic temperatures. The sentence has been deleted in this revised MS.

[References]

1. Lee, Cheng Chi. "Is human hibernation possible?." *Annu. Rev. Med.* 59 (2008): 177-186.
2. Dugbartey, G. J. *et al.* A hibernation-like state for transplantable organs: is hydrogen sulfide therapy the future of organ preservation? *Antioxidants & Redox Signaling* 28, 1503-1515 (2018).

8. Line 88 "Stresses" should not be capitalized.

Response: This mistake has been corrected (**Line 81**).

--Experimental procedures

9. Why were only male bats used?

Response: We have addressed the reason why only male bats were chosen in **Lines 133-134**.

Main Text: Lines 133-134

Because female bats are often pregnant during hibernation season, only adult male bats were used.

♥ Please allow us to explain a little bit more:

In the Fangshan cave, most female bats (Rickett's big-footed bat, family Vespertilionidae) are pregnant during hibernation season (> 70%, the empirical experience from our team workers). Using female bats represents a higher possibility that we sample "more bats" than we really need, which is not fit ethical regulation of bat conservation. Moreover, it is difficult for us to quickly identify those non-pregnant female bats in the field during bat sampling in cold winter. Since the physiological condition of male bats is comparatively simpler than that in female bats, we therefore used only male bats in this study. References below show general reproduction patterns of hibernating bats.

[References]

1. Female reproductive patterns in hibernating bats. BA Oxberry - Reproduction, 1979 - rep.bioscientifica.com
2. Jonasson, K. A. & Willis, C. K. Changes in body condition of hibernating bats support the thrifty female hypothesis and predict consequences for populations with white-nose syndrome. *PLoS one* 6, e21061 (2011).
3. Willis, C. K., Brigham, R. M. & Geiser, F. Deep, prolonged torpor by pregnant, free-ranging bats. *Naturwissenschaften* 93, 80-83 (2006)

10. Are 2 h versus 24 h aroused bats fully active and awake? Some animals which "arouse" in the middle of hibernation are not actually active – while their T_b is returned to normal they are still not moving around.

Response: It is true that sometimes when T_b of the hibernated animals is returned to normal they are still not moving around. In this study, however, both bats (2 h and 24 h aroused

bats) were separately and individually placed in the cloth bag, and all bags were put together in a big backpack. We carried the backpack and transported it ~50 min by mountain hiking to the lab beneath the mountain.

(<http://www.bio.bris.ac.uk/research/bats/China%20bats/Research%20Centre.htm>)

We observed that 2 h aroused bats are in fact not fully active, they don't wave their wings and seem to be torpid, but are responsive to human disturbances. Interestingly, those 24 h aroused bats are more active and sneaky. They often move and try to fly out from the bags, so that we have to tight up the bags and pay attention. We think the 24 h aroused bats is far more active than 2 h aroused bats, and they can be "quick enough" to escape their constraint.

We also noted that the protein expression pattern of 2 h aroused bats is more like that of torpid bats (Fig. 1C and S6). Although, the 24 h aroused bats is more active than 2 h aroused bats, but they are not exactly the same as summer active bats (Fig. 3C). We may consider these 24 h aroused bats are in a state of un-natural activation (artificial arousal).

11. Why were 11-week old mice used? In general, would be good to outline the relevance of including mice and how when compared to hibernation experimental conditions, it offers a different perspective, even though this is fairly obvious.

Response: We couldn't agree more! We have addressed these points in **Lines 145** (Method section) and **Lines 585-589** (Discussion section).

Main Text: Lines 145

Mice of this age were chosen as their torpor-like states had been successfully induced³¹⁻³³.

Main Text: Lines 585-589

As mammalian hibernators undergo multiple bouts of torpor and arousal but show no significant organ damage after hibernation^{1,75}, they are excellent models for application of natural protective mechanisms in humans^{76,77}. Fasted and cold-induced non-hibernating laboratory mice that mimic torpor-like conditions of hibernators are used to explore the molecular differences between induced and natural torpor³¹⁻³³.

[References]

1. Drew, K. L., Rice, M. E., Kuhn, T. B. & Smith, M. A. Neuroprotective adaptations in hibernation: therapeutic implications for ischemia-reperfusion, traumatic brain injury and neurodegenerative diseases. *Free Radical Biology and Medicine* 31, 563-573 (2001).

75. Carey, H. V., Andrews, M. T. & Martin, S. L. Mammalian hibernation: cellular and molecular responses to depressed metabolism and low temperature. *Physiological reviews* 83, 1153-1181 (2003).

76. Bouma, H. R. et al. Induction of torpor: mimicking natural metabolic suppression for biomedical applications. *Journal of cellular physiology* 227, 1285-1290 (2012).

77. Cerri, M. et al. Hibernation for space travel: Impact on radioprotection. *Life sciences in space research* 11, 1-9 (2016).

31. Uchida, Y., Tokizawa, K. & Nagashima, K. Characteristics of activated neurons in the suprachiasmatic nucleus when mice become hypothermic during fasting and cold exposure. *Neuroscience letters* 579, 177-182 (2014).

32. Sato, N., Marui, S., Ozaki, M. & Nagashima, K. Cold exposure and/or fasting modulate the relationship between sleep and body temperature rhythms in mice. *Physiology & behavior* 149, 69-75 (2015).

33. Tokizawa, K., Uchida, Y. & Nagashima, K. Thermoregulation in the cold changes depending on the time of day and feeding condition: physiological and anatomical analyses of involved circadian mechanisms. *Neuroscience* 164, 1377-1386 (2009).

12. What was the body temperature of the mice monitored?

Response: The body temperature of mice were measured as described in Lines 149-150 (Method section).

Main Text: Lines 149-150

The body weight and food intake of mice were recorded by electronic scale, and rectal temperature of mice were recorded by thermometer (Center 309 data logger) as shown in Fig. S1 and supplement Table S1.

Fig. S1

Supplement Table S1

Mice & groups	Body weight (g) Rectal temperature (°C)					Other indexes Food Consumption (g)
	Day 1	Day 3	Day 5	Day 7	Day 9	
Control 1	28.4 35.1	30.3 36.4	30.5 36.5	31.6 35.5	31.5 34.9	13.9
Control 2	26.2 35.0	26.8 36.5	27.5 36.9	28.0 35.6	27.8 35.6	13.7
Control 3	27.6 36.9	28.2 35.6	29.1 36.8	29.7 36.9	29.4 36.5	13.6
Control 4	29.3 36.0	29.9 36.4	30.3 36.9	31.0 37.3	30.4 36.8	14.1
Fast 1	27.8 37.0	29.1 36.2	29.3 37.4	31.0 36.1	22.7 31.5	0
Fast 2	27.7 35.9	29.1 35.1	29.4 36.0	29.9 37.4	23.7 30.6	0
Fast 3	28.4 37.3	28.8 37.1	29.4 35.4	30.1 35.5	29.5 29.7	0
Fast 4	26.7 36.0	27.5 35.2	27.9 36.4	28.6 36.1	27.9 30.1	0
Cold 1	27.1 37.3	28.2 36.7	29.5 35.5	29.3 36.6	30.0 33.0	17.5
Cold 2	28.9 36.4	29.9 35.5	30.6 35.4	31.6 37.2	32.0 32.9	17.6
Cold 3	28.4 36.2	29.2 37.2	29.8 36.7	30.9 36.9	31.3 32.8	18.0
Cold 4	29.2 36.1	30.3 36.3	31.0 37.1	31.8 37.3	32.4 31.6	17.2

13. What controls were performed to ensure the purity of the mitochondrial fraction?

Response: In this study, the mitochondrial fraction prepared from Mitochondria Isolation Kit for Tissue (89801, Thermo Scientific, USA). We have added Fig. S2 to show how the purity of the fractions was defined (Lines 162-164).

Main Text: Lines 162-164

Several mitochondrial protein markers (e.g., NADH-ubiquinone oxidoreductase 75 kDa subunit and glutamate dehydrogenase 1) were checked to ensure the purity of mitochondrial fractions (Fig. S2).

Fig. S2

♥ Please allow us to explain a little bit more:

The purity of mitochondrial fractions prepared from liver of *Myotis ricketti* bats (at four different physiological states), *Cynopterus sphinx* bats, pig, mouse, and rat were checked by the Western blotting using four mitochondrial protein markers, including glutamate dehydrogenase 1 (**GLUD1**, mitochondrion matrix), ornithine carbamoyl-transferase (**OTC**, mitochondrion matrix), carbamoyl-phosphate synthase (**CPS1**, mitochondria and nucleus), mitochondrial; and NADH-ubiquinone oxidoreductase 75 kDa subunit (**NDUSF1**, mitochondrial inner membrane) and one non-mitochondrial protein marker, beta-actin, cytoplasmic 1 (**ACTB**, cytoplasm and nucleus).

Different species were used to test the capability of the Mitochondria Isolation Kit for Tissue (89801, Thermo Scientific, USA) and we found the kit works well on liver tissues from different mammalian species as **ACTB** signals were barely detected in most of the species. The signals of **NDUSF1** and **OTC** were not found in *Cynopterus sphinx* bats, it is possible that these antibodies did not react with protein peptides of this bat species.

Furthermore, results of Fig. 1A showed 228 proteins were identified only in liver-derived mitochondrial fraction but not in liver fraction, and 381 proteins were identified only in liver fraction but not in mitochondrial fraction, which would support the successful enrichment of the liver mitochondria using the kit.

14. Why did the authors choose SDS-PAGE to fractionate proteins prior to mass spec, instead of other methods (e.g. on column methods)? How does recovery of proteins from gels compare to other methods.

Response: We have addressed the main reason in the manuscript (**Lines 179-180**).

Main Text: Lines 179-180

This gel-based protein fractionation and recovery method was used because all the techniques involved are well-established in our lab.

♥ Please allow us to explain a little bit more here:

Compared to gel-free techniques, gel-based method is inexpensive, powerful, simple and easy to use in fractionation of complex protein mixtures, removal of interfering contaminants, assessment of sample amount and complexity using in-gel total protein staining. For label-free quantification of proteins¹, one common challenge could be poor

recovery of proteins from in-gel digestion as compared to that from in-solution digestion². However, in this study, the number of identified proteins (>1000 proteins in liver fraction and >800 proteins in mitochondrial fraction, Fig. 1A) indicated that our method of protein fractionation should be work out fine.

[References]

1. Cox, Jürgen, et al. "Accurate proteome-wide label-free quantification by delayed normalization and maximal peptide ratio extraction, termed MaxLFQ." *Molecular & cellular proteomics* 13.9 (2014): 2513-2526.
2. Jafari, Mohieddin, et al. "Comparison of in-gel protein separation techniques commonly used for fractionation in mass spectrometry-based proteomic profiling." *Electrophoresis* 33.16 (2012): 2516-2526.

15. During Co-IP how efficient is the elution process? Does any antibody remain bound to its antigen?

Response: We have re-addressed the elution process in **Line 251-253**.

Main Text: Lines 251-253

The bead-Ab-Ag complexes were washed three times with washing buffer, eluted with elution buffer (50 mM Tris-HCl, pH 6.8, 10% glycerol, 1.25% β-mercaptoethanol, 2% SDS, 0.1% bromophenol blue), and heated at 100°C for 10 min.

♥ Allow us to explain a little bit more:

In this study, we followed the denaturing elution protocol of the kit, so that the antibody and antigen are eluted by elution buffer and together with the heating process would warrant elution of most proteins containing antibody itself from the beads.

--Results

16. Could consider labeling the different parts of Figure 1B with i, ii, iii – this way you don't have to refer to it in the text by "upper right". I would also consider adding the title "subcellular, molecular function, and biological process" inside its respective grey box. Similar comments for other figures.

Response: We appreciate very much for this suggestion and have revised them accordingly. The changes were made in **Lines 282, 286, 403-404, 410, 457, 458, 462, and 471-472, and Fig. 1B, 4A, 5C, and 5D**.

--Discussion

17. Many of the proteins measured are followed by a superscript – e.g. see line 519 which lists a superscript number for each protein listed. What are these in reference to? These are easily confused with references.

Response: We have changed the Arabic numeral-only superscript to become the superscript form of ***italic bold alphabet (a or b)*** and ***Arabic numerals***, for example in main text lines 506-507: ...EEF2⁵⁵, GRP78⁰¹, PDI^{23|37|41}, PPIA⁵⁰, PPIB³⁶, UBC⁶⁹, and PSMB6⁶⁵ (Fig. 3A),... were changed to become ...EEF2^{a55}, GRP78^{a01}, PDI^{a23|a37|a41}, PPIA^{a50}, PPIB^{a36}, UBC^{a69}, and PSMB6^{a65} (Fig. 3A),....

♥ **Please allow us to explain a little bit more:**

In this revised manuscript, we have tried our best to re-label these proteins identified (Fig. 3 and Tables S3 and S4). If this remains not good enough to let readers distinguish the reference mark from the identified protein, then please allow us to own the chance to delete all superscripts.

18. Since the amount of data and discussion is so dense, it would be good to introduce figure 6 earlier and use it as a guide for the reader as you describe results.

Response: We have described our findings correlated to all signals in advance (Fig. 6). See **Lines 522-524**.

Main Text: Lines 522-524

This possibility is supported by our finding that Nrf2, Akt, and NF-κB were coordinately activated via the PERK-EIF2-ATF4 signaling pathway in torpid *Myotis ricketti* bats (Fig. 6).

Reviewer #2 (Remarks to the Author):

--General comments

*This manuscript describes a significant proteomics study examining factors involved in stress in hibernating and active bats. Although the findings are interesting, they are not especially novel since stress proteins have been found to increase in a number of hibernators. I was disappointed that the authors did not cite some of these studies (by **Hannah Carey, Sandy Martin, Frank van Breukelen, and others**) in the discussion. These labs have studied **stress proteins, NF-κB, apoptosis and elongation factors** in hibernators and published several papers each. The lack of citations in this instance, especially given the similarities in the current authors' findings seems strange. Please do a thorough review of the hibernation literature and discuss the similarities between these studies and the current one.*

Response: Agree a lot! We have carefully reviewed the hibernation literature maintained by the reviewer and cited those papers correlated closely to our findings in the main text as follows (**Lines 110-111, 491-492, 544-545, and 604-606**):

Main text:

Introduction, Lines 110-111

A nearly 20-fold increase in Akt abundance and activation of NF-κB were seen in the gut of hibernating squirrels^{28,29}.

Discussion, Lines 491-492

Selective binding of 4E-binding protein-1 (4E-BP1) to EIF4E to initiate protein translation was found in golden-mantled ground squirrels during winter hibernation⁵⁵⁻⁵⁷.

Discussion, Lines 544-545

A previous study showed that apoptosis in hibernating 13-lined ground squirrels was suppressed due to a 12-fold increase in the expression of Bcl-xL⁵⁶.

Discussion, Lines 604-606

As mammalian hibernation is a complex physiological adjustment against cold and adverse surroundings, further studies using other mammals are required to learn how signal cross talks are achieved to adapt to harsh environments ⁷⁸.

♥ Allow us to explain a little:

We consider a series of hibernation studies done by Hannah Carey, Sandy Martin, and Frank van Breukelen and others were tremendous, we have been cited several of their papers in our previous publications. We try hard to make up this careless mistake and hope this revised manuscript is reasonable and acceptable in its limit of 5000 words (introduction, Results, and Discussion), if not, please allow one more chance for us to improve ourselves.

--Advice

Beyond this general comment, I have some suggestions for revisions that should be addressed:

19. Throughout the manuscript, the bat is referred to by its scientific name only. What is the common name for this animal? Please include this information at least once.

Response: We have added the common name (Rickett's big-footed bat) of *Myotis ricketti* bats in **Line 114**.

20. For the immunoblotting, how much protein was loaded for each type of lysate (in ug)? Also, I'm not sure why the developed band was normalized to the Ponceau band of the same size. This seems like an odd way to normalize the data. There could be a lot of other proteins near the one of interest that affect the Ponceau staining. This could skew the results significantly.

Response: The experimental details for **loading**, separation gel, blocking, primary antibody condition, second antibody condition, washing time and exposure time are listed on **Table S6**. Besides, we have been carefully revised the statement of this method in **Lines 240-243**.

Main text: Lines 240-243

The relative quantity of a protein was determined by dividing the density value of the protein band on Western blot by the total density value of all protein bands in a lane of a Ponceau stained blot ³⁹ (Fig. S3).

♥ Allow us to explain a little:

As the expression level of many proteins was significantly changed in tropid bats compared to active bats, so that this method of normalization is relatively effective, accurate, and it seems acceptable¹.

[Reference]

39. Romero-Calvo, I. et al. Reversible Ponceau staining as a loading control alternative to actin in Western blots. *Analytical biochemistry* 401, 318-320 (2010).

1. Xue, Huiling, et al. "Molecular signatures and functional analysis of beige adipocytes induced from in vivo intra-abdominal adipocytes." *Science advances* 4.7 (2018): eaar5319.

21. The manuscript needs a stronger rationale for the mouse experiments. Why were they done? Certainly, there are enough studies out there about cold exposure in mice and everyone can agree that cold exposure in mice is not the same as torpor in a hibernator. I don't really see what these experiments add beyond saying this same thing.

Response: We have clarified our rationale to do the mouse experiment in **Lines 145, 585-589**.

Main text:

Lines 145

Mice of this age were chosen as their torpor-like states had been successfully induced³¹⁻³³.

Discussion, Lines 585-589

As mammalian hibernators undergo multiple bouts of torpor and arousal but show no significant organ damage after hibernation^{1,75}, they are excellent models for application of natural protective mechanisms in humans^{76,77}. Fasted and cold-induced non-hibernating laboratory mice that mimic torpor-like conditions of hibernators are used to explore the molecular differences between induced and natural torpor³¹⁻³³.

[References]

1. Drew, K. L., Rice, M. E., Kuhn, T. B. & Smith, M. A. Neuroprotective adaptations in hibernation: therapeutic implications for ischemia-reperfusion, traumatic brain injury and neurodegenerative diseases. *Free Radical Biology and Medicine* 31, 563-573 (2001).

75. Carey, H. V., Andrews, M. T. & Martin, S. L. Mammalian hibernation: cellular and molecular responses to depressed metabolism and low temperature. *Physiological reviews* 83, 1153-1181 (2003).

76. Bouma, H. R. et al. Induction of torpor: mimicking natural metabolic suppression for biomedical applications. *Journal of cellular physiology* 227, 1285-1290 (2012).

77. Cerri, M. et al. Hibernation for space travel: Impact on radioprotection. *Life sciences in space research* 11, 1-9 (2016).

31. Uchida, Y., Tokizawa, K. & Nagashima, K. Characteristics of activated neurons in the suprachiasmatic nucleus when mice become hypothermic during fasting and cold exposure. *Neuroscience letters* 579, 177-182 (2014).

32. Sato, N., Marui, S., Ozaki, M. & Nagashima, K. Cold exposure and/or fasting modulate the relationship between sleep and body temperature rhythms in mice. *Physiology & behavior* 149, 69-75 (2015).

33. Tokizawa, K., Uchida, Y. & Nagashima, K. Thermoregulation in the cold changes depending on the time of day and feeding condition: physiological and anatomical analyses of involved circadian mechanisms. *Neuroscience* 164, 1377-1386 (2009).

♥ Allow us to explain a little:

Indeed, we found mice may be not like small hibernators that have a real torpor, but we still try to figure out the differences between an induced torpor and a real torpor. We believe the more we know the more we can do to induce a real torpor in a mammalian non-hibernator; although the hope is frail, but it is hard to give up.

22. At the top of page 10 (line 448) the authors state: "To investigate whether anti-apoptosis in torpid bats was mediated by Erk1/2 signaling..." The experiments done do not show any sort of mediation, however. To do this, one would have to block Erk1/2 signaling and show that the effect goes away. This should be reworded.

Response: We have rewritten the sentence in **Lines 432-433** to become: To investigate whether anti-apoptotic response in torpid bats was correlated with Erk1/2 activation^{9,53}, the total amounts of Erk1/2 and phosphorylated Erk1/2 were determined.

23. Also, on page 13 (line 513), the authors state that “...we have determined for the first time the cross talk among Nrf2, Akt and NF-κB signaling...”. Although the IP assays did show interaction between some of these proteins, I think that this is too strong a statement. The study did not really determine the cross talk but it did show the possibility.

Response: We have rewritten and deleted the imprecise words according to the reviewer's comments in **Lines 599-600**.

Main Text: Lines 599-600

In conclusion, we have determined for the first time **coordinated activation of Nrf2, Akt, and NF-κB signaling pathways** via the PERK-EIF2-ATF4 **regulatory axis** in torpid bats (Fig. 6)

Again, we enormously thank these constructive comments from both professional reviewers and hope this revised manuscript can reach the criteria of publication in Communications Biology.

**Regards,
Jennifer Pan and Wenjie Huang**

REVIEWERS' COMMENTS:

Reviewer #1 (Remarks to the Author):

The authors have adequately addressed all of my comments and I approve publication. Thank you.

Reviewer #3 (Remarks to the Author):

The Authors have appropriately responded to all of my initial comments. I approve of the manuscript in its current form.